

# The tropical tropopause layer in reanalysis data sets

Susann Tegtmeier[1], James Anstey[2], Sean Davis[3], Rossana Dragani[4], Yayoi Harada[5], Ioana Ivanciu[1], Robin Pilch Kedzierski[1], Kirstin Krüger[6], Bernard Legras[7], Craig Long[8], James S. Wang[9], Krzysztof Wargan[10,11], and Jonathon S. Wright[12]

[1]GEOMAR Helmholtz Centre for Ocean Research Kiel, 24105 Kiel, Germany

[2]Canadian Centre for Climate Modelling and Analysis, ECCC, Victoria, Canada

[3]Earth System Research Laboratory, National Oceanic and Atmospheric Administration, Boulder, CO 80305, USA

[4]European Centre for Medium-Range Weather Forecasts, Reading, RG2 9AX, UK

[5]Japan Meteorological Agency, Tokyo, 100-8122, Japan

[6] Section for Meteorology and Oceanography, Department of Geosciences, University of Oslo, 0315 Oslo, Norway

[7]Laboratoire de Météorologie Dynamique, CNRS/PSL-ENS, Sorbonne University Ecole Polytechnique, France

[8]Climate Prediction Center, National Centers for Environmental Prediction, National Oceanic and Atmospheric Administration, College Park, MD 20740, USA

[9]Institute for Advanced Sustainability Studies, Potsdam, Germany

[10]Science Systems and Applications, Inc., Lanham, MD 20706, USA

[11]Global Modeling and Assimilation Office, Code 610.1, NASA Goddard Space Flight Center, Greenbelt, MD 20771, USA

[12]Department of Earth System Science, Tsinghua University, Beijing, 100084, China





**Abstract**
The tropical tropopause layer (TTL) is the transition region between the well mixed, convective
troposphere and the radiatively controlled stratosphere with air masses showing chemical and
dynamical properties of both regions. The representation of the TTL in meteorological
reanalysis data sets is important for studying the complex interactions of circulation,
convection, trace gases, clouds and radiation. In this paper, we present the evaluation of TTL
characteristics in reanalysis data sets that has been performed as part of the SPARC
(Stratosphere– troposphere Processes and their Role in Climate) Reanalysis Intercomparison
Project (S-RIP).
The most recent atmospheric reanalysis data sets all provide realistic representations of the
major characteristics of the temperature structure within the TTL. There is good agreement
between reanalysis estimates of tropical mean temperatures and radio occultation data, with
relatively small cold biases for most data sets. Temperatures at the cold point and lapse rate
tropopause levels, on the other hand, show warm biases in reanalyses when compared to
observations. This tropopause-level warm bias is related to the vertical resolution of the
reanalysis data, with the smallest bias found for data sets with the highest vertical resolution
around the tropopause. Differences of the cold point temperature maximise over equatorial
Africa, related to Kelvin wave activity and associated disturbances in TTL temperatures. Model
simulations of air mass transport into the stratosphere driven by reanalyses with a warm cold
point bias can be expected to have too little dehydration.
Interannual variability in reanalysis temperatures is best constrained in the upper TTL, with
larger differences at levels below the cold point. The reanalyses reproduce the temperature
responses to major dynamical and radiative signals such as volcanic eruptions and the QBO.
Long-term reanalysis trends in temperature in the upper TTL show good agreement with trends
derived from adjusted radiosonde data sets indicating significant stratospheric cooling of
around –0.5 to –1 K/decade. At 100 hPa and the cold point, most of the reanalyses suggest small
but significant cooling trends of –0.3 to –0.6 K/decade that are statistically consistent with
trends based on the adjusted radiosonde data sets.
Advances of the reanalysis and observational systems over the last decades have led to a clear
improvement of the TTL reanalyses products over time. Biases of the temperature profiles and
differences in interannual variability clearly decreased in 2006, when densely sampled radio
occultation data started being assimilated by the reanalyses. While there is an overall good
agreement, different reanalyses offer different advantages in the TTL such as realistic profile
and cold point temperature, continuous time series or a realistic representation of signals of
interannual variability. Their use in model simulations and in comparisons with climate model
output should be tailored to their specific strengths and weaknesses.







## 1. Introduction

The tropical tropopause layer (TTL) is the transition region between the well-mixed, convective troposphere and the radiatively-controlled stratosphere. The vertical range of the TTL extends from the region of strong convective outflow near 12-14 km to the highest altitudes reached by convective overshooting events, around 18 km (Highwood and Hoskins, 1998; Folkins et al 1999; Fueglistaler et al., 2009; Randel and Jensen, 2013). Air masses in the TTL show dynamical and chemical properties of both the troposphere and the stratosphere, and are controlled by numerous processes on a wide range of length and time scales. Complex interactions among circulation, convection, trace gases, clouds and radiation in the TTL make this region a key player in radiative forcing and chemistry-climate coupling. As the TTL is the main gateway for air entering the stratosphere, stratospheric chemistry and composition, and especially the abundances of ozone, water vapour and aerosols, are strongly impacted by the properties of air near the tropical tropopause.

The tropopause is the most important physical boundary within the TTL, serving to separate the turbulent, moist troposphere from the stable, dry stratosphere. The position of the tropopause is determined by the thermal properties of the TTL, as a negative, tropospheric vertical temperature gradient changes into a positive stratospheric temperature gradient. The role of the tropopause as a physical boundary is evident not only from the vertical temperature structure, but also from the distributions of atmospheric trace gases and clouds.

In the tropics, two definitions of the tropopause are widely used: one based on the cold point and one based on the characteristics of the lapse rate. The cold point tropopause is defined as the level at which the vertical temperature profile reaches its minimum (Highwood and Hoskins, 1998) and air parcels en route from the troposphere to the stratosphere encounter the lowest temperatures. Final dehydration typically occurs at these lowest temperatures, so that the cold point tropopause effectively controls the overall water vapour content of the lower stratosphere (Randel et al., 2004a) and explains its variability (Fueglistaler et al., 2009). While the cold point tropopause is an important boundary in the tropics where upwelling predominates, this definition of the tropopause is irrelevant for water vapor transport into the stratosphere at higher latitudes. The lapse rate tropopause, on the other hand, offers a globally-applicable definition of the tropopause, marking a vertical discontinuity in the static stability. The lapse rate tropopause is defined as the lowest level at which the lapse rate decreases to 2 K km$^{-1}$ or less, provided that the average lapse rate between this level and all higher levels within 2 km does not exceed 2 K km$^{-1}$ (World Meteorological Organization, 1957). The tropical lapse rate tropopause is typically ~0.5 km (~10 hPa) lower and ~1 K warmer than the cold point tropopause (Seidel et al., 2001).

Over recent decades, the thermal characteristics of the TTL and tropopause have been obtained from tropical radiosonde and Global Navigation Satellite System - Radio Occultation (GNSS-RO) upper air measurements. Radiosonde profiles offer temperature, wind and air pressure data at a high vertical resolution. However, climate records based on radiosonde data often suffer from spatial inhomogeneities or time-varying biases due to changes in instruments and measurement practices (Seidel and Randel, 2006; Wang et al., 2012). Climate records from



radio occultation data offer much better spatial coverage and density, but are only available
starting from 2002. As a result, studies of long-term variability and trends in TTL and
tropopause properties have also used reanalysis data (e.g., Santer et al., 2003; Gettelman et al.,
2010, Xie et al., 2014).
Meteorological reanalysis data sets are widely used in scientific studies of atmospheric
processes and variability, either as initial conditions for historical model runs or in comparisons
with climate model output. Often, they are utilized as "stand-ins" for observations, when the
available measurements lack the spatial or temporal coverage needed. Each atmospheric
reanalysis system consists of a fixed global forecast model and assimilation scheme. The system
combines short-range forecasts of the atmospheric state with available observations to produce
best-guess, consistent estimates of atmospheric variables such as temperatures and winds.
Spurious changes in the reanalysis fields can arise from changes in the quality and quantity of
the observations used as input data, which complicates the analysis of variability and trends.
Further discontinuities in reanalysis-based time series can originate from the joining together
of distinct execution streams (Fujiwara et al., 2017).
Among the various TTL characteristics such as composition, radiation budgets and cloud
properties, the vertical temperature structure and the position and temperature of the cold point
are of particular importance for transport and composition studies. Many off-line chemistry-
transport models or Lagrangian particle dispersion models are driven by reanalysis data sets
(e.g., Chipperfield, 1999; Krüger et al., 2009; Schoeberl et al, 2012). Their representation of the
cold point determines how realistically such models simulate dehydration and stratospheric
entrainment processes. Process studies of TTL dynamics such as equatorial wave variability are
also often based on the TTL temperature structure in reanalysis data sets (e.g., Fujiwara et al.,
2012). Finally, reanalysis cold point temperature and height have been used in the past for
comparison to model results and in investigations of long-term changes (e.g., Gettelman et al.,
2010). Information on the quality and biases of TTL temperature and tropopause data are
important for all above listed studies of transport, composition, dynamics and long-term
changes of the TTL.
A comparison of the reanalysis products available at the end of the 1990s (including ERA-15,
ERA-40 and NCEP-NCAR R1) with other climatological datasets showed notable differences
in temperatures near the tropical tropopause (Randel et al., 2004b). While the ECWMF
reanalyses agreed relatively well with radiosonde observations at 100 hPa, NCEP-NCAR R1
showed a warm bias of up to 3 K, probably resulting from low vertical resolution and the use
of poorly-resolved satellite temperature retrievals (Fujiwara et al., 2017). Comparisons of
winter temperatures at 100 hPa between more recent reanalyses, such as MERRA, NCEP CFSR
and ERA-Interim, and Singapore radiosonde observations show better agreement, with
reanalyses generally 1-2 K too cold at this level (Schoeberl et al, 2012). While many studies
have highlighted the characteristics of individual reanalysis data sets, a comprehensive
intercomparison of the TTL among all major atmospheric reanalyses is currently missing.
Here, we investigate whether reanalysis data sets reproduce key characteristics of the
temperature and tropopauses in the TTL. This work has been conducted as part of the SPARC



(Stratosphere–troposphere Processes and their Role in Climate) Reanalysis Intercomparison Project (S-RIP) (Fujiwara et al., 2017) and presents some of the key findings from the S-RIP report Chapter 8 on the TTL. Climatologies of the tropical cold point and lapse rate tropopauses as derived from modern reanalysis data sets are compared to high-resolution radio occultation data (Section 3). We also investigate temporal variability and long-term changes in tropopause levels and temperature within the TTL (Section 4). The observational and reanalysis data sets used in the evaluation are introduced in Section 2, and a discussion and summary of the results are provided in Section 5.

## 2 Data and methods

### 2.1 Observational data sets

High-resolution observations of the TTL are available from tropical radiosonde stations. However, climate records of radiosonde temperature, height and pressure data often suffer from inhomogeneities or time-varying biases due to changes in instruments or measurement practices (Seidel and Randel, 2006). Adjusted radiosonde temperature data sets at 100 hPa, 70 hPa and corresponding trends at the cold point have been created by removing such inhomogeneities (Wang et al., 2012, and references therein). In this chapter, we use several independently adjusted radiosonde data sets, including RATPAC (Free et al., 2005), RAOBCORE (Haimberger, 2007) and HadAT (Thorne et al., 2005) as well as the unadjusted, quality-controlled radiosonde data set IGRA (Durre et al., 2006) covering the S-RIP core time period (1980–2010).

Since 2002, high-resolution temperature and pressure data in the TTL are also available from satellite retrievals based on the GNSS-RO technique. Recent studies have demonstrated good agreement between GNSS-RO and radiosonde temperature profiles (e.g. Anthes et al., 2008; Ho et al., 2017). We use a zonal mean data set constructed from measurements collected by the Challenging Minisat Payload (CHAMP, Wickert et al., 2001), Gravity Recovery and Climate Experiment (GRACE, Beyerle et al., 2005), Constellation Observing System for Meteorology, Ionosphere, and Climate (COSMIC, Anthes et al., 2008), Metop-A (von Engeln et al., 2011), Metop-B, Satélite de Aplicaciones Científicas-C/Scientific Application Satellite-C (SAC-C, Hajj et al., 2004), and TerraSAR-X (Beyerle et al., 2011) missions. All data are re-processed or post-processed occultation profiles with moisture information ('wetPrf' product) as provided by the COSMIC Data Analysis and Archive Center (CDAAC, https://cdaac-www.cosmic.ucar.edu/cdaac/products.html). Observational temperature records at reanalysis model levels in the TTL region have been determined by interpolating GNSS-RO temperature profiles with the barometric formula, taking into account the lapse rate between levels. For each profile, the cold point and lapse rate tropopause characteristics were identified based on the cold point and WMO criteria, respectively.

We also use a daily data set of cold point temperatures obtained from all GNSS-RO missions, gridded on a 5°x5° grid between 30°N and 30°S. For each 5° wide latitude band, we apply a two-dimensional fast Fourier transform to detect Kelvin wave anomalies for planetary wavenumbers 1–15, periods of 4–30 days and equivalent depths of 6–600 following the theoretical dispersion curves for Kelvin waves as in Wheeler and Kiladis (1999). We allow a



wider range of equivalent depths, since it has been shown that Kelvin waves tend to propagate faster around the tropical tropopause than they do in the troposphere (Kim and Son, 2012). The filtered anomalies represent cold point temperature variability that propagates in the same wavenumber-frequency domain as Kelvin waves, i.e. when the temperature is modulated by Kelvin waves present around the tropopause. The spatial variance of the filtered signals is used to calculate a monthly index as a measure of the amount of Kelvin wave activity in the TTL. The index is calculated as the $1\sigma$ standard deviation over the filtered anomalies at all spatial grid points. Time periods of enhanced Kelvin wave activity are defined as the months when the index is larger than the long-term mean plus the $1\sigma$ standard deviation of the whole time series. Based on this definition, we determined 20% of all months to be characterized by enhanced Kelvin wave activity.

## 2.2 Reanalysis data sets

We evaluate eight "full-input" reanalyses, defined as systems that assimilate surface and upper-air conventional and satellite data. In this paper, we focus on ERA-Interim (Dee et al., 2011), ERA5 (Hersbach et al., 2018), JRA-25 (Onogi et al., 2007), JRA-55 (Kobayashi et al., 2015), MERRA (Rienecker et al., 2011), MERRA-2 (Gelaro et al., 2017), NCEP-NCAR Reanalysis 1 (Kistler et al., 2001; referred to hereafter as R1), and CFSR (Saha et al., 2010). We limit our analyses to the S-RIP core intercomparison period 1980–2010. Due to availability at the time of the evaluations, ERA5 is only evaluated over 2002–2010. Details of each reanalysis, including model characteristics, physical parameterizations, assimilated observations, execution streams, and assimilation strategies have been summarized by Fujiwara et al. (2017). MERRA-2 data access was through the Global Modeling and Assimilation Office (GMAO, 2015).

Global temperature fields in the reanalysis data sets are produced by assimilating conventional (surface and balloon), aircraft, and satellite observations. The most important sources of assimilated data for stratospheric temperatures are the microwave and infrared satellite sounders of the TOVS suite (1979–2006) and the ATOVS suite (1998–present). All of the above reanalysis systems assimilate microwave and infrared radiances from these instruments, except for NCEP-NCAR R1 which assimilates temperature retrievals instead. Measurements from the ATOVS suite, which has a higher number of channels, have been assimilated from about 1998, although the exact start dates differ among the reanalyses. The introduction of ATOVS considerably improved the vertical resolution of the assimilated data. Some of the reanalyses (ERA-Interim, ERA5, MERRA, MERRA-2, and CFSR) also assimilate radiances from the hyperspectral infrared sounders AIRS (2002–present), IASI (2008–present), and/or CrIS (2012–present), although the latter was not available for assimilation during the intercomparison period considered here. Radiance biases associated with instrument changes, inaccurate calibration offsets, orbital drifts or long-term $CO_2$ changes can propagate into the reanalysis fields (e.g. Rienecker et al., 2011).

All full-input reanalyses assimilate upper-air temperature observations from radiosondes which are available at a very high vertical resolution. Systematic errors in radiosonde profiles caused by effects of solar radiative heating on the temperature sensor (Nash et al., 2011) have typically



been corrected either onsite or at the reanalysis centre before assimilation (Fujiwara et al.,
2017). In order to avoid discontinuities or inconsistencies in temperature time series from
radiosondes, several reanalysis systems use homogenized temperature data sets such as
RAOBCORE (ERA-Interim, JRA-55, MERRA, MERRA-2) and RICH (ERA5). Earlier
reanalyses (ERA-40 and JRA-25) used simplified homogenization approaches that mostly
corrected for daily and seasonal variations. Although the detailed quality control procedures for
radiosonde and other conventional data imported from the global distribution network can vary
among the individual reanalyses, the conventional data archives are often shared among the
centres (see also Fujiwara et al., 2017).
Recent reanalysis systems have also included information from GNSS-RO instruments by
assimilating observations of the bending angle up to 30 km (Cucurull et al., 2013). Assimilating
these high vertical resolution data affects reanalysis temperature and provides an additional
'anchor' for adaptive bias correction of satellite radiances. JRA-55 assimilates refractivity
profiles up to 30 km, which are functions of temperature, humidity, and pressure. For all recent
reanalyses, the advent of the COSMIC mission in 2006 significantly increased the number of
GNSS-RO profiles available for assimilation. Details of the various GNSS-RO data assimilated
by ERA5, ERA-Interim, JRA-55, MERRA-2 and CFSR up to the end of 2010 are listed in
**Table 1**.

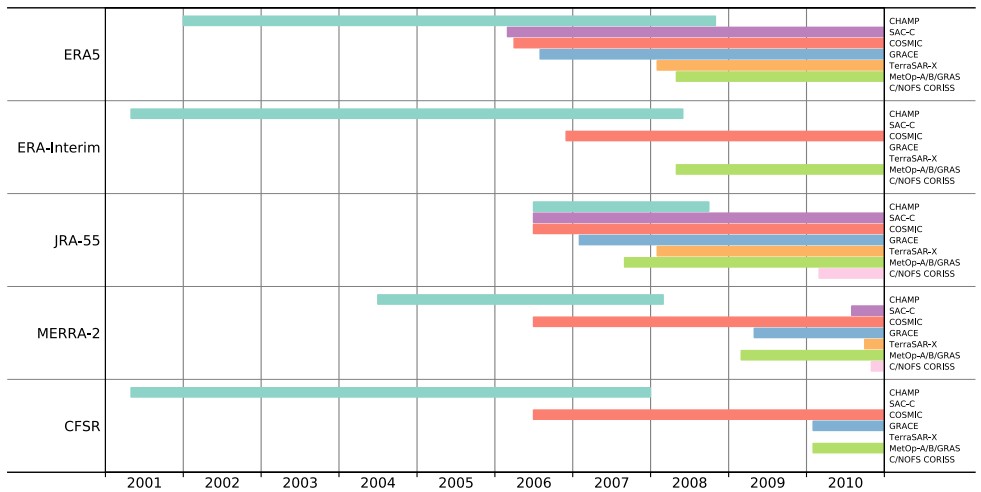

**Table 1**. List of GNSS-RO data assimilated by the reanalysis systems with starting dates prior to the end
of 2010.
Among the observational data sets, radiosonde and GNSS-RO data are our best source of
information about the TTL. While the reanalyses assimilate versions of these data, it is not
automatic that they reproduce the data exactly. For instance, discrepancies exist between
reanalysis stratospheric temperatures and those derived from their radiance input data (Long et
al., 2017). In fact, it is a subject of ongoing research how well reanalyses fit the data they
assimilate (Simmons et al., 2014, Wright and Hindley 2018). In general, the data assimilation
methodology relies on an interplay among a vast number of diverse observations, model





simulations and bias correction schemes. The degraded vertical resolution of the reanalyses,
compared to radiosonde and GNSS-RO data also leads to differences, especially for derived
quantities such as the tropopause location and temperature, which will be investigated in the
following evaluations.
The reanalysis models resolve the TTL with different vertical resolutions, as illustrated in **Fig.**
**1**. The number of model levels between 200 and 70 hPa varies among the reanalyses from a
low of 4 (NCEP-NCAR R1) to a high of 21 (ERA5), corresponding to vertical resolutions
between ~1.5 km and ~0.2 km. In addition to the native model levels, all reanalyses provide
post-processed data on standard pressure levels with at least four levels situated between 200
and 70 hPa (**Fig. 1**). We show that reanalysis-based estimates of tropopause temperature,
pressure and height compare much better to observations when they are derived from model-
level data than when they are derived from pressure-level data (Section 3.1). Another sensitivity
study demonstrates that tropopause temperatures directly calculated from monthly-mean fields
have a warm bias of 0.5 K compared to tropopause temperatures based on 6-hourly data (not
shown here). Therefore, we derive the cold point and lapse rate tropopause characteristics for
each reanalysis using model-level data at each grid point at 6-hourly temporal resolution. Zonal
and long-term averages are then calculated by averaging over all grid points, and represent the
final step of data processing.

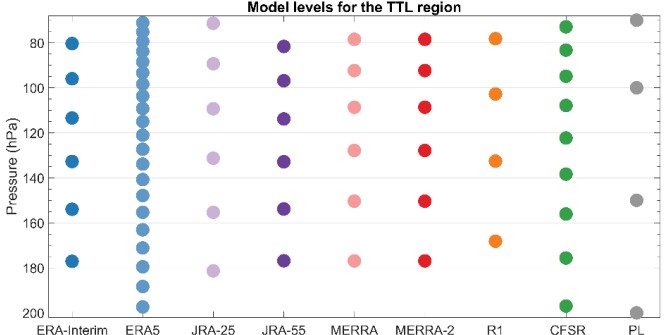

**Figure 1.** Model-level pressure values for different reanalysis data sets in the TTL using a fixed surface
pressure of 1013.25 hPa. Standard pressure levels (PL) in the TTL region are also shown.
**2.3 Methods**
Given the strong gradients of temperature and static stability in the TTL, the vertical resolution
of the reanalysis data sets is an important factor in cold point and lapse rate tropopause
calculations. For each reanalysis, tropopause heights and temperatures can be derived either
from model- or pressure-level data (**Fig 1**). A comparison of the CFSR cold point tropopause
based on model- and pressure-level temperature data is shown here to demonstrate the clear
advantage of the finer model-level resolution (**Fig. 2**). The cold point tropopause from CFSR
model-level data for the time period 2002–2010 agrees well with radio occultation results, with
differences of less than 1.5 K and 0.2 km at all latitudes. The tropopause derived from CFSR





pressure-level data, on the other hand, shows larger differences. This estimate is up to 0.4 km
too low and up to 3 K too warm, illustrating the need to use data with high vertical resolution
to identify and describe the tropopause. The following climatological tropopause comparisons
are all based on model-level data.

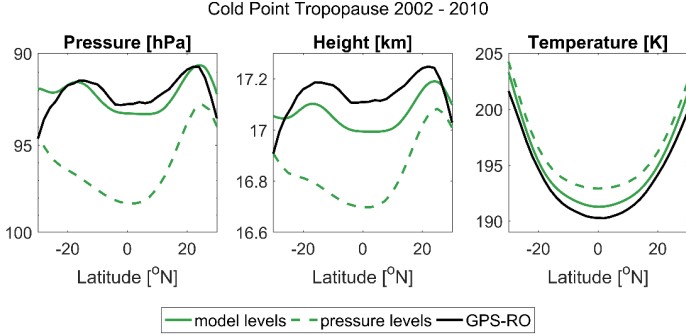

**Figure 2.** Latitudinal distributions of zonal-mean cold point tropopause pressure (left), altitude (centre)
and temperature (right) based on radio occultation data (black) and CFSR model-level (green solid) and
pressure-level (green dashed) data during 2002–2010.
The evaluation of the interannual variability (Section 4) is based on time series of
deseasonalized monthly temperature, pressure and altitude anomalies calculated relative to the
mean annual cycle during 2002–2010. To study variability driven by tropospheric and
stratospheric forcing, we identify and isolate the variations based on a standard multivariate
regression analysis:
$T(t) = A_1 \cdot \mathrm{QBO1}(t) + A_2 \cdot \mathrm{QBO2}(t) + B \cdot \mathrm{ENSO}(t) + D \cdot \mathrm{VOL}(t).$   (1)
Here $\mathrm{QBO1}(t)$ and $\mathrm{QBO2}(t)$ are orthogonal time series representing QBO variations constructed
as the first two EOFs of the Freie Universität Berlin (FUB) radiosonde stratospheric winds
(Naujokat, 1986). $\mathrm{ENSO}(t)$ is the multivariate ENSO index
(https://www.esrl.noaa.gov/psd/enso/mei/) and $\mathrm{VOL}(t)$ is the stratospheric aerosol optical depth
from the Global Space-based Stratospheric Aerosol Climatology (Thomason et al., 2018). The
standard error of the regression coefficients was derived based on the bootstrap method. The
QBO temperature amplitude is calculated as the difference between the averaged maxima and
averaged minima values of the time series of the QBO temperature variations $A_1 \cdot \mathrm{QBO1}(t) +$
$A_2 \cdot \mathrm{QBO2}(t)$.
The long-term trends of the reanalyses temperature time series have been derived as the
regression coefficient of a linear function that provides the best fit in a least-squares sense. The
trend error bars are as the standard error of the slope with an effective sample size. Significance
is tested based on two-tailed test with a 95% confidence interval.





## 3 Temperature and tropopause characteristics

Tropical mean temperatures from reanalyses at two standard pressure levels (100 hPa and 70 hPa) and at the two tropopause levels are compared to radio occultation data for the time period 2002–2010 (**Fig. 3**). At 100 hPa, reanalysis temperatures agree well with radio occultation data with differences between –0.35 K (too cold; ERA-Interim and ERA5) and 0.43 K (too warm; CFSR). At 70 hPa, the agreement is even better, with differences ranging from –0.29 K (JRA-55) to 0.12 K (JRA-25). However, nearly all reanalyses show warm biases at both tropopause levels, with differences of up to 1.2 K compared to the observations. Most likely, the excess warmth of tropopause estimates based on reanalysis products stems from the limited vertical resolution of the reanalysis models in the TTL region. The best agreement is found for the reanalysis with the highest vertical resolution (ERA5; 0.05 K too warm at the cold point tropopause). The data set with the lowest vertical resolution (NCEP-NCAR R1) is 2.2 K too warm, outside the range displayed in Figure 3.

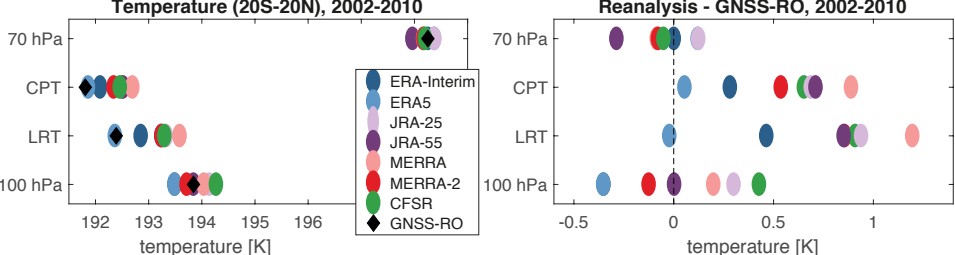

**Figure 3.** Tropical mean (20°S-20°N) temperatures at 100 hPa, the lapse rate tropopause (LRT), the cold point tropopause (CPT) and 70 hPa from reanalyses and GNSS-RO data during 2002–2010 (left panel). Differences between the GNSS-RO and reanalysis temperatures are shown in the right panel. At 100 hPa, ERA-Interim is hidden by ERA-5, at the LRT, MERRA-2 is hidden by JRA-55, and at 70 hPa, ERA5 is hidden by JRA-25 and MERRA is hidden by MERRA-2.

Temperature profile comparisons between 140 and 70 hPa at the native model level resolution have been conducted for the five most recent reanalyses (ERA5, ERA-Interim, JRA-55, MERRA-2, CFSR). All reanalyses tend to be colder than the observations in the tropical mean (**Fig. 4**), but differences are relatively small and the agreement is good overall. CFSR and ERA5 agree best with the radio occultation data with mean biases of around –0.06 K and –0.28 K, respectively, averaged over the whole vertical range. ERA-Interim and MERRA-2 agree very well at upper levels but show large deviations near 100 hPa (ERA-Interim; –0.82 K) and below 110 hPa (MERRA-2; –0.67 K), respectively. The evaluation demonstrates that temperature comparisons at standard pressure levels (**Fig. 3**) can be biased by up to 0.5 K, with CFSR showing a positive bias (0.45 K) at the 100 hPa standard pressure level but very good agreement (-0.05 K) at nearby native model levels. Such biases can result from vertical interpolation of temperature data in regions with large lapse rate changes.





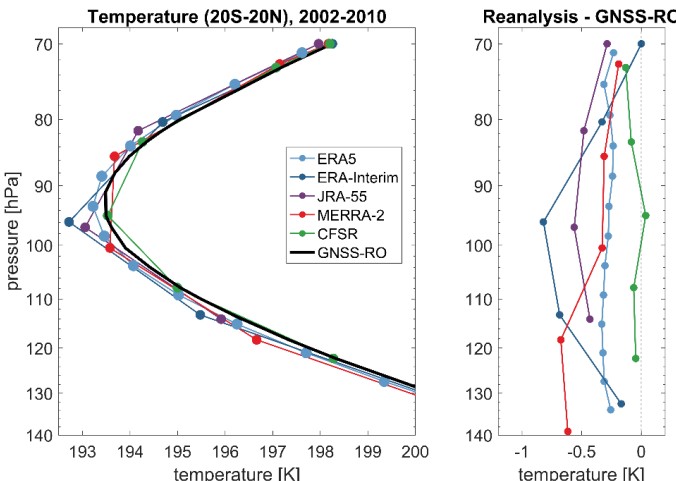

**Figure 4** Tropical mean (20°S–20°N) temperature profiles at reanalysis model levels between 140 and 70 hPa (left panel) during 2002–2010 and differences between reanalysis and GNSS-RO temperatures (right panel).

Comparing the temperature profiles to the tropopause values (**Fig. 3 and 4**) reveals that despite the five reanalyses having negative biases at model levels, they mostly have positive biases at the cold point and lapse rate tropopause levels. As the discrete values corresponding to reanalysis model levels are unable to reproduce the true minimum temperature as recorded in a near-continuous profile, this difference is expected for the cold point tropopause. Similarly, the lapse rate tropopause criteria might typically be fulfilled at lower levels for data at coarser resolution, thus resulting in a warm bias at the lapse rate tropopause on average. Overall, our results indicate that the negative temperature bias at model levels is more than cancelled out by the positive bias introduced when calculating the cold point and lapse rate tropopauses. Linking the temperature profile and tropopause comparisons, this 'bias shift' is about 0.3 K for ERA5, 0.6 K for CFSR and 1 K or larger for ERA-Interim, MERRA-2 and JRA-55. In consequence, ERA5, with both a small negative bias at the model levels and a small bias shift provides the most realistic tropopause temperatures. CFSR also has a relatively small bias shift, but the relatively unbiased temperature profile does not permit any error cancelation via this shift, so that cold point and lapse rate tropopause levels based on CFSR are systematically too warm.

Agreement between the reanalysis temperature profiles and GNSS-RO data clearly improves when the comparison is restricted to the 2007–2010 time period, when the more densely-sampled COSMIC data were assimilated (**Table 1**). This point is illustrated by comparison of temperature time series from reanalyses and observations at two model and both tropopause levels (**Fig. 5**). For ERA5, ERA-Interim and MERRA-2, the cold bias with respect to GNSS-RO at model levels decreases after 2007, most likely because of the high number of daily COSMIC profiles available for assimilation from this time onwards. Cold biases at model levels are accompanied by warm biases in the tropopause temperatures, which, for ERA-Interim and ERA5, increase after 2007. Here, the advantage of a reduced temperature bias at model levels





comes at the expense of an increased temperature bias at the tropopause. CFSR and MERRA-
2 show no such systematic change of their tropopause temperatures over time when compared
to GNSS-RO data. JRA-55 is the only reanalysis product for which cold point and lapse rate
tropopause temperatures agree slightly better with GNSS-RO estimates after 2007.

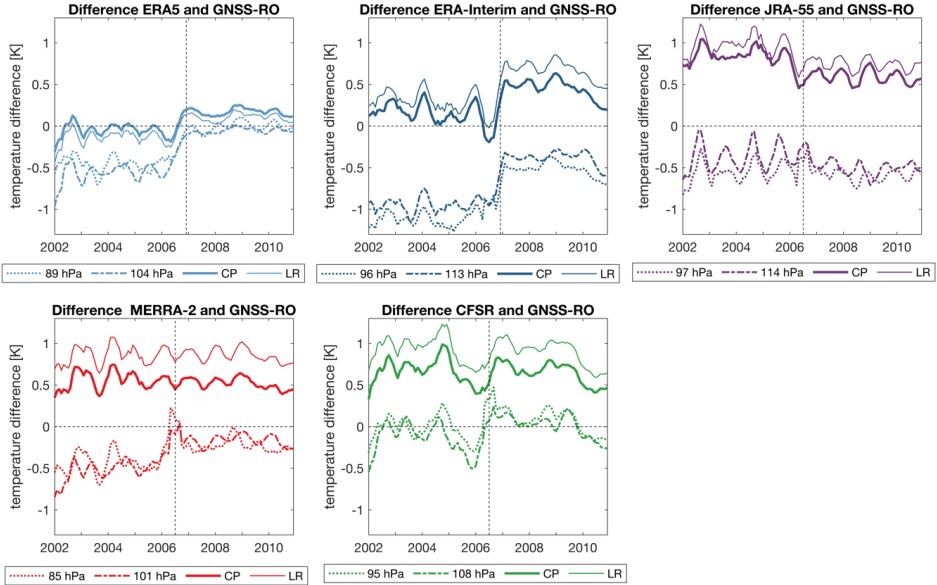

**Figure 5.** Tropical mean (20°S-20°N) time series of temperature differences between reanalysis and
radio occultation at the cold point (CP) and lapse rate (LR) tropopause levels, as well as selected
reanalysis model levels. Vertical lines indicate when the assimilation of COSMIC radio occultation data
started.
Evaluations of the latitudinal structure of the cold point tropopause for 2002–2010 are based on
comparisons to radio occultation data (**Fig. 6**). All reanalysis data sets produce tropopause
levels that are too low and too warm, with the latter related to vertical resolution as explained
above. The observations show that average cold point temperatures are lowest right around the
equator. The reanalyses fail to reproduce this latitudinal gradient, indicating more constant cold
point temperatures across the inner tropics between 10°S and 10°N with a less pronounced
minimum at the equator. As a consequence, the largest differences in cold point tropopause
temperatures relative to GNSS-RO data are at the equator and the best agreement is around
20°S/20°N for all reanalysis data sets.





The cold point altitude and pressure exhibit little north–south variability, ranging from 16.9 km
(94 hPa) to 17.2 km (91.8 hPa). With respect to the seasonal cycle, it is well known that the
temperature and altitude of the cold point tropopause are linked, with the coldest temperatures
and highest altitudes observed during boreal winter (e.g., Seidel et al., 2001; Kim and Son,
2012). This relationship does not hold with respect to the zonal mean: the highest cold point
altitudes are located around 20°S/20°N, while the lowest cold point temperatures are located
near the equator. The higher altitude/lower pressure of the cold point tropopause around
20°S/20°N results from zonally-variable features linked to tropospheric pressure regimes, such
as particularly low tropopause pressures over the Tibetan plateau during boreal summer (Kim
and Son, 2012). The reanalysis data sets capture most of this latitudinal structure, showing
roughly constant differences between about 0.1 and 0.2 km (0–2 hPa). The largest differences
are found for NCEP-NCAR R1 in the Southern Hemisphere, where the cold point tropopause
based on R1 is both higher and warmer than observed. The best agreement with respect to cold
point temperatures is found for ERA5 and ERA-Interim, which are around 0.2 K and 0.4 K
warmer than the radio occultation data, respectively. All other reanalysis data sets are in close
agreement with each other, with differences from the observations of between 0.5 K and 1 K.
The altitude and pressure of the cold point tropopause are captured best by ERA5, CFSR,
MERRA, MERRA-2 and JRA-55, which all produce cold point tropopauses that are slightly
too low (~0.1 km). ERA-Interim, despite very good agreement in cold point temperature, shows
slightly larger biases in cold point altitude (~0.2 km) relative to the GNSS-RO benchmark.

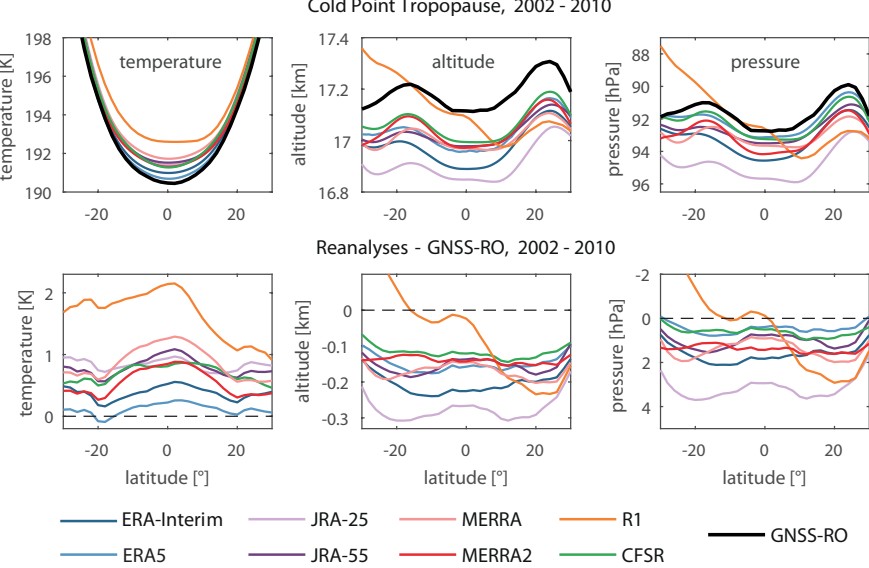

**Figure 6.** Latitudinal distributions of zonal-mean cold point tropopause temperature (left), altitude
(centre) and pressure (right) based on radio occultation data and reanalysis products during 2002–2010
(upper row). Differences between radio occultation and reanalysis estimates are shown in the lower row.



We investigate the temperature biases and their maxima near the equator by analysing latitude–
longitude variations in the cold point tropopause relative to GNSS-RO estimates for four of the
reanalyses (**Fig. 7**). To show differences at relatively high spatial resolution, we focus on the
period 2007–2010. A wealth of observational studies has shown that the coldest tropopause
temperatures are located over the Maritime continent and the West Pacific (Highwood and
Hoskins, 1998), with secondary minima over equatorial South America and Africa coinciding
with other centres of deep convective activity (Gettelman et al., 2002). The colocation of
tropospheric convective activity with zonal asymmetries in cold point temperature can be
explained by the radiative cooling effects of cirrus clouds overlying deep convection (Hartmann
et al., 2001) or diabatic cooling associated with convective detrainment (Sherwood et al., 2003).
Furthermore, it has been suggested that the response of equatorial waves to convective heating
influences the structure of the cold point tropopause (Kim and Son, 2012; Nishimoto and
Shiotani, 2012; Nishimoto and Shiotani, 2013). The dominant wave modes responsible for cold
point temperature variability are linked to equatorial Kelvin waves and the Madden-Julian
oscillation.

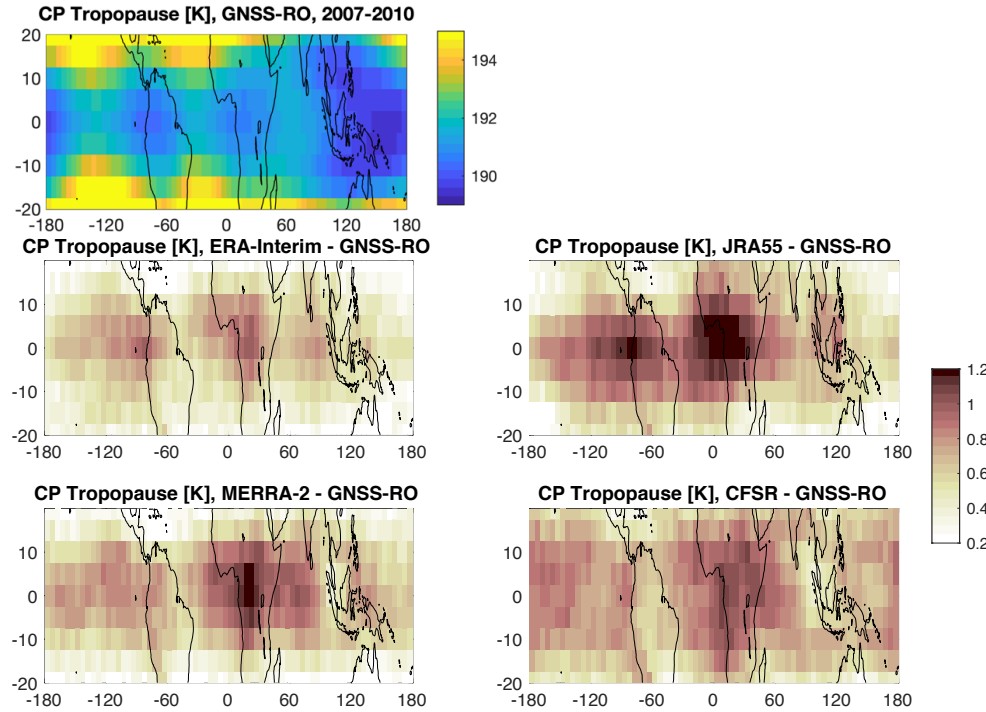

20  **Figure 7.** Latitude–longitude distributions of annual mean GNSS-RO cold point temperatures (upper
21  left) and differences between cold point temperatures from individual reanalyses and those from GNSS-
22  RO during 2007–2010 (lower panels).





For the analysed reanalyses (ERA-Interim, MERRA2, JRA55, and CFSR), differences with
respect to the observations are largest in the inner tropics over central Africa, reaching values
50% to 100% greater than the zonal mean differences. This region is characterized by a local
cold point minimum that results from deep convection and its interaction with equatorial waves.
There is also evidence of a secondary maximum in the differences over equatorial South
America or the East Pacific, although the magnitude and location of this maximum differ among
the reanalyses.
The convective centre over the Western Pacific warm pool, where the cold point tropopause is
coldest, does not show enhanced biases relative to the observations. One possible explanation
for the bias distribution might link the enhanced temperature differences to Kelvin wave activity
that maximizes over Central Africa but is weaker over the West Pacific (Kim et al., 2019). As
the Kelvin waves disturb the temperature profile at small vertical scales, the reanalyses may be
particularly unsuited to estimating cold point temperatures in regions of strong Kelvin wave
activity. We average cold point temperatures from reanalyses and observations over time
periods of enhanced Kelvin wave activity. For CFSR, composite differences for periods with
enhanced wave activity are compared in **Fig. 8** to mean differences averaged over the whole
2007–2010 period. While mean biases over Central Africa are less than 1 K, average differences
during periods of enhanced Kelvin wave activity are as large as 1.4 K. The same is true for
other reanalyses (not shown here), with the exception of ERA-Interim, suggesting that in most
cases Kelvin waves contribute to the spatial structure of biases in cold point tropopause
estimates based on reanalysis products.

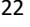
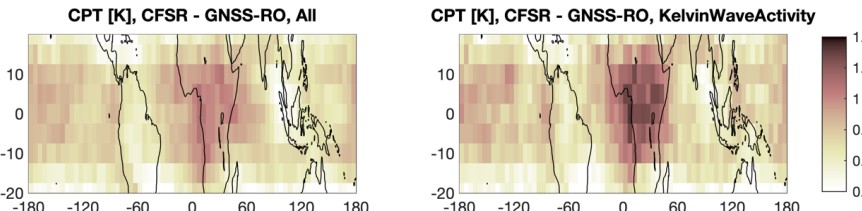

**Figure 8.** Latitudinal-longitude sections of the differences between GNSS-RO and CFSR cold point
temperatures for 2007-2010 (left panel) and for time periods of high wave activity (right panel).
The zonal mean lapse rate tropopause (**Fig. 9**) at the equator is found at similar temperatures
and heights as the cold point tropopause, being only slightly warmer and lower. Poleward of
10°S/10°N, however, the lapse rate tropopause height decreases considerably faster than the
cold point height, since the cold point is more often located at the top of the inversion layer
while the lapse rate tropopause is located at the bottom of the inversion layer (Seidel et al.,
2001). Lapse rate tropopause temperatures based on reanalysis data are on average about 0.2 K
to 1.5 K too warm when compared to radio occultation data (see **Fig. 3** and associated
discussion) with best agreement for ERA5 and ERA-Interim. Consistent with this temperature
bias, lapse rate tropopause levels based on reanalysis data are about 0.2 km to 0.4 km lower





than those based on radio occultation data. The latitudinal structure of lapse rate tropopause
temperatures reveals slightly larger biases at the equator and better agreement between 10°–20°
in each hemisphere, and is generally very similar to the latitudinal distribution of biases in cold
point temperatures (**Fig. 6**). The altitude of the lapse rate tropopause shows considerable zonal
variability, ranging from 14.5 km to 16.7 km. All reanalyses capture the plateau in lapse rate
tropopause altitudes between 20°S and 20°N and the steep gradients in these altitudes on the
poleward edges of the tropics.

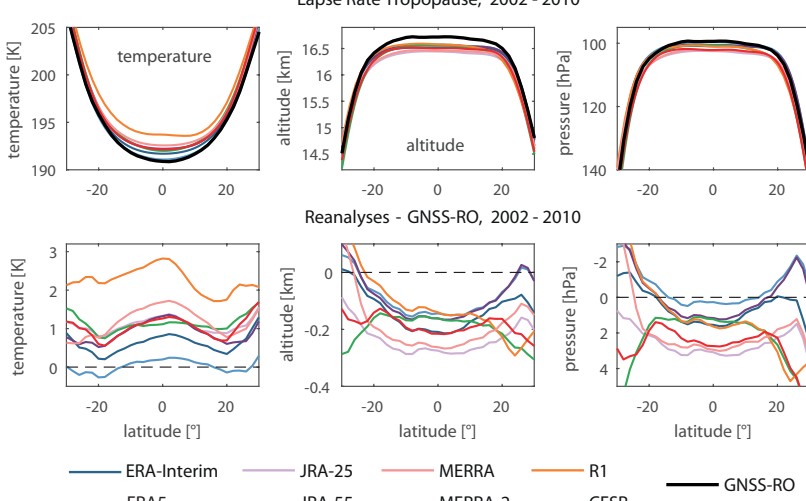

**Figure 9.** Latitudinal distributions of zonal-mean lapse rate tropopause temperature (left), altitude
(centre) and pressure (right) based on radio occultation data and reanalysis products during 2002–2010
(upper row). Differences between radio occultation and reanalysis estimates are shown in the lower row.



## 4 Interannual variability and long-term changes

It has long been recognized that inter-annual variations in TTL temperatures are strongly affected by both tropospheric (e.g. ENSO) and stratospheric (e.g. QBO, solar, volcanic) variability (Randel et al., 2000; Zhou et al., 2001, Krüger et al 2008). Time series of deseasonalized monthly 70 hPa temperature anomalies and cold point temperature, pressure and altitude anomalies are shown in **Fig. 10**. Anomalies are calculated relative to the mean annual cycle during 2002–2010 for each dataset. The performance of the reanalyses with respect to both the spread among reanalyses and their agreement with observations is much better at the 70 hPa level than at the cold point level. The older reanalyses NCEP-NCAR R1 and JRA-25 generally show larger deviations from the RAOBCORE time series. The level of agreement among the reanalyses and between reanalyses and observations improves over time, with a step-like improvement around 1998–1999 that is likely associated with the TOVS-to-ATOVS transition. The higher vertical resolution of measurements from the ATOVS suite (see, e.g. Figure 7 in Fujiwara et al., 2017) is known to reduce differences among the reanalysis with respect to stratospheric temperature (Long et al., 2017) and polar diagnostics (Lawrence et al., 2018). Within the TTL, temperature biases improve from values of 1–2 K to around 0.5 K following the TOVS-to-ATOVS transition. This agreement improves further after 2002, when many of the more recent reanalyses started assimilating AIRS and GNSS-RO data (**Table 1**; see also Figure 8 in Fujiwara et al., 2017).

Interannual variability at 70 hPa is dominated by the stratospheric QBO signal, which is reproduced by all reanalysis data sets. The amplitudes of the QBO temperature variations in all datasets based on a multilinear regression analyses over 1980-2010 are shown in **Fig. 11**. At 70 hPa, the observational radiosonde data sets give QBO variations of 2.1–2.2 K. Reanalyses agree well with the observations and show QBO variations of 2–2.4 K. The only exception is NCEP-NCAR R1, which clearly underestimates the signal compared to radiosondes and other reanalyses, with an amplitude of 1.7 K. Best agreement with the radiosonde data sets is found for MERRA2, MERRA and CFSR. The influence of ENSO on TTL temperatures shows large longitudinal variations with positive anomalies over the Maritime Continent and West Pacific and negative anomalies over the East Pacific. While the zonally resolved response patterns agree well between observations and reanalyses (not shown here), the zonal mean responses are not significant (not shown here). Positive temperature anomalies following the eruptions of El Chichón in 1982 and Mount Pinatubo in 1991 can be detected in **Fig. 10** for all reanalyses, consistent with the results of Fujiwara et al. (2015).

At the cold point, NCEP-NCAR R1 is a clear outlier, with much warmer temperature anomalies than any other data set during the period prior to 2005. However, differences among the more recent reanalyses are also relatively large, with ERA-Interim (on the lower side) and CFSR (on the upper side) showing differences as large as 2 K in the early years of the comparison. Given that existing homogenized radiosonde data sets also show deviations of up to 1.5 K at this level (Figure 2 in Wang et al., 2012), we cannot deduce which reanalysis data set is most realistic. Note that the radiosonde time series from IGRA shown here should not be used for evaluating long-term changes (see Wang et al., 2012 for details), but only for assessing the representation





of interannual variability. Periods of particularly pronounced interannual variability alternate
with relatively quiescent ones. The QBO temperature signal at the cold point is weaker than at
70 hPa but still well captured by all of the reanalysis data sets except for NCEP-NCAR R1. The
amplitude of interannual variability is also smaller at the cold point than at 70 hPa, but with
larger month-to-month variations (not shown here).

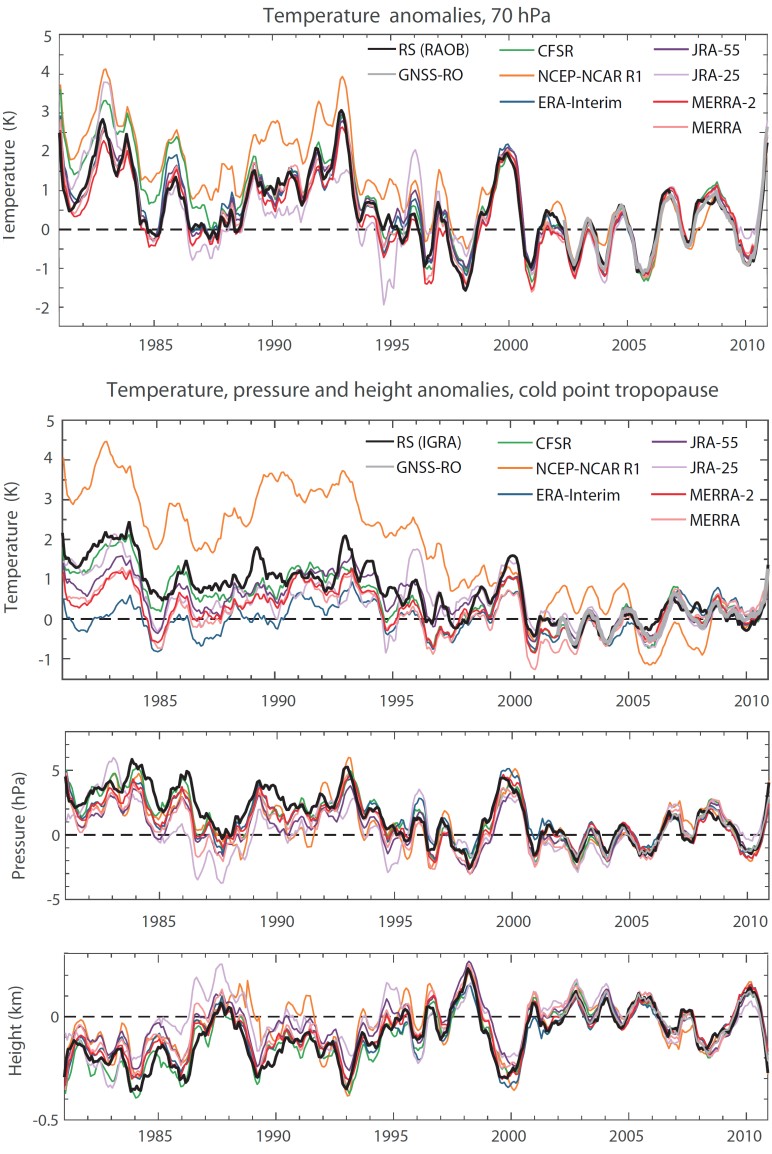

**Figure 10.** Time series of deseasonalized anomalies in 70 hPa temperature (upper), cold point
temperature (upper middle), cold point pressure (lower middle) and cold point altitude (lower) averaged
over the tropics (20°S–20°N) and evaluated relative to the reference period 2002–2010. Time series are





shown for reanalysis products, radiosonde data (RAOBCORE and IGRA) and radio occultation data
(GNSS-RO). Time series are smoothed with a 7-months running mean.
Interannual variability in cold point pressure and altitude shows better agreement among the
data sets than that in cold point or 70 hPa temperature. During the first 15 years of the record,
the reanalysis cold point tropopause levels are mostly shifted toward higher altitudes and lower
pressures, consistent with lower temperatures during this period. Anomalies in cold point
temperature are in most cases matched by anomalies in cold point pressure and altitude, with a
warmer cold-point temperature (e.g. around 1999–2000) corresponding to lower tropopause
(negative altitude anomaly and positive pressure anomaly) and vice versa. The older reanalyses
NCEP-NCAR R1 and JRA-25 again show the largest overall differences. The agreement
improves over time, with the most consistent results found for the period after 2002.

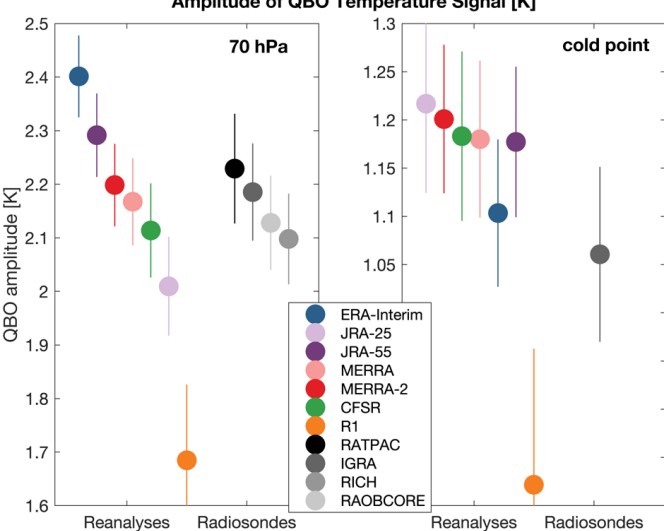

**Figure 11.** Amplitude of QBO temperature signal for 10°S-10°N at 70 hPa and the cold point derived
from a multilinear regression analyses for radiosonde and reanalysis data sets.
Long-term temperature changes are evaluated over the 1979-2005 time period due to the
availability of adjusted tropopause trends from radiosonde data sets (see Wang et al., 2012 for
details). Both radiosonde records suggest significant cooling at the 70 hPa level (**Fig. 12**).
Trends derived from reanalysis data can be problematic due to changes in the assimilated
observations. Given this potential limitation, it is of interest to examine whether the reanalysis
trends are consistent with the hypothetically more reliable trends derived from homogenized
observational records. At 70 hPa, temperature trends based on the reanalysis data sets span
almost exactly the same range (–0.5 to –1.1 K/decade) as those based on the radiosonde data



sets (–0.5 to –1 K/decade). All reanalysis- and observationally-based trends are significant at
this level, confirming the stratospheric cooling reported by many previous studies (e.g., Randel
et al., 2009). Satellite data from the Microwave Sounding Unit channel 4 (~13–22 km) suggests
smaller trends of around –0.25 K/decade over 1979–2005 (Maycock et al., 2018) or –0.4
K/decade over 1979–2009 (Emanuel et al., 2013). However, the much broader altitude range
of this MSU channel includes both stratospheric and tropospheric levels, which impedes a direct
comparison with trends at 70 hPa.
At the 100 hPa and cold point levels, the situation is completely different. The available
adjusted radiosonde data sets show in some cases uncertainties larger than the respective
temperature trends at these levels. Only a few of the available data sets indicate a statistically
significant cooling based on a methodology that adjusts the cold point trend to account for
nearby fixed pressure-level data and day–night differences (Wang et al., 2012). Based on the
trends shown in Wang et al. (2012) for five adjusted radiosonde data sets, we show here the
smallest and largest reported trends and consider their range (including the reported error bars)
as the observational uncertainty range. Similar to the observations, the reanalysis data sets
suggest a large range in cold point temperature trends, from no trend at all (0 K/decade for
ERA-Interim) to a strong cooling of –1.3 K/decade (NCEP-NCAR R1). The latter is outside of
the observational uncertainty range and can thus be considered unrealistic. All other reanalyses
suggest small but significant cooling trends of –0.3 K/decade to –0.6 K/decade. JRA-25, JRA-
55, MERRA, and MERRA-2 agree particularly well and produce trends in the middle of the
observational uncertainty range. Overall, due to the large uncertainties in radiosonde-derived
cold point temperature trends, all reanalyses except for R1 are statistically consistent with at
least one of the observational data sets.

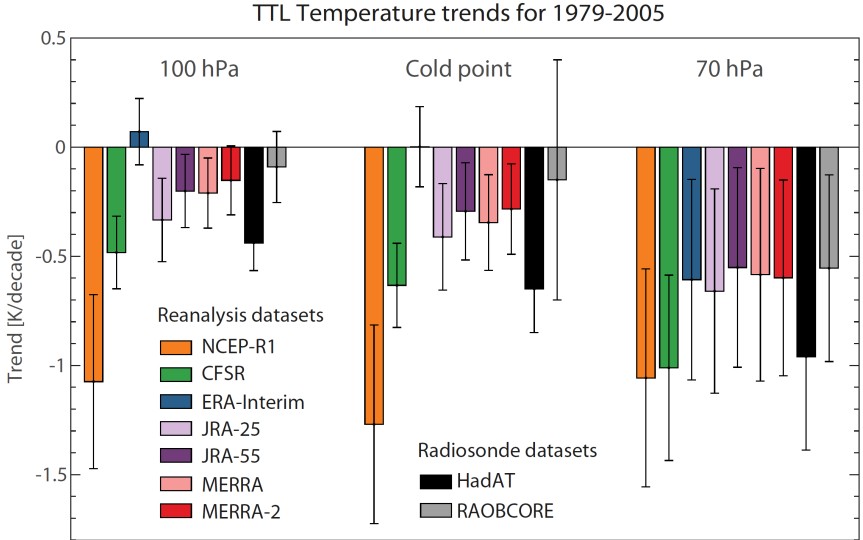





**Figure 12**. Linear trends in tropical mean (20°S–20°N) temperature (K/decade) at 100 hPa, the cold
point and 70 hPa for the time period 1979-2005. Error bars indicate ±2σ uncertainty in the trend and
account for serial auto-correlation.
Temperature trends at 100 hPa are very similar to trends at the cold point level, and again
suggest consistency among most of the reanalysis and radiosonde data sets, with the notable
exception of R1. Nearly all data sets suggest slightly smaller cooling trends (–0.15 K/decade to
-0.5 K/decade) relative to the cold point consistent with the fact that the cold point is at slightly
higher altitudes than 100 hPa. Among the data sets, only ERA-Interim produces a warming
trend (0.07 K/decade), although this result is not statistically significant.
**5 Summary**
Meteorological reanalyses are widely used in scientific studies of TTL processes being utilizied
as ''stand in observations'' or for driving transport models. The most recent atmospheric
reanalysis data sets (ERA5, ERA-Interim, MERRA-2, JRA-55, and CFSR) all provide realistic
representations of the major characteristics of temperature structure within the TTL. There is
good agreement between reanalysis estimates of tropical mean temperatures between 140 and
70 hPa and GNSS-RO retrievals, with relatively small cold biases for most data sets. CFSR
shows the best agreement with GNSS-RO in this layer with a mean bias of –0.06 K. Agreement
between the temperature profiles and the GNSS-RO data clearly improves when the comparison
is restricted to the period after 2007, when the densely-sampled COSMIC data were assimilated
by all reanalyses.
Temperatures at the cold point and lapse rate tropopause levels show warm biases in reanalyses
when compared to observations. This tropopause-level warm bias is opposite to the cold bias
found at all model levels and is most likely related to difficulties in determining the true cold
point and lapse rate tropopause levels from discrete temperature profiles with coarse vertical
resolution. Our analysis confirms that the magnitude of the bias shift is consistent with the
vertical resolution of the reanalysis data, with the smallest bias shifts found for data sets with
the highest vertical resolution around the tropopause (ERA5 and CFSR). The negative
temperature bias at model levels is often cancelled out by the positive bias introduced when
identifying the lapse rate and cold point tropopause locations. As a result, ERA5, which has a
small negative bias at model levels, has the most realistic tropopause temperatures, while CFSR,
which produces the most realistic model-level temperature profile, has a warm bias of 0.6–0.9
K at the cold point and lapse rate tropopause levels. Older reanalyses like MERRA, JRA-25
and especially NCEP-NCAR R1 show the largest temperature biases at the tropopause levels.
The zonal structure of tropopause temperature reveals that the biases in reanalysis relative to
observations maximise at or near the equator. All of the recent reanalyses produce a realistic
horizontal structure of cold point temperature with minima corresponding to the centres of
tropical deep convection. Differences between reanalyses and observations are greatest over
equatorial Africa. These enhanced differences are possibly related to Kelvin wave activity and
associated disturbances in TTL temperatures that also maximize in this region.



Interannual variability in reanalysis temperatures is best constrained in the upper TTL (70 hPa), with larger differences at lower levels such as the cold point and 100 hPa. The reanalyses reproduce the temperature responses to major dynamical and radiative signals such as volcanic eruptions and the QBO. Agreement among the reanalyses and between the reanalyses and observations generally improves over time, with a step-like improvement around the TOVS-to-ATOVS transition in 1998–1999 and in 2006 with the beginning of assimilation of COSMIC GNSS-RO data. Interannual variability is lower at the cold point and 100 hPa relative to 70 hPa, but with larger month-to-month fluctuations causing larger discrepancies among the reanalyses. As at 70 hPa, NCEP-NCAR R1 is a clear outlier. Interannual variability in cold point pressure and altitude shows better agreement than that in TTL temperature. Anomalies in cold point temperatures are in most cases matched by corresponding anomalies in cold point pressure and altitude.

Long-term reanalysis trends in temperature at 70 hPa show good agreement with trends derived from adjusted radiosonde data sets. All reanalyses and observational data sets indicate significant stratospheric cooling at this level of around –0.5 K/decade to –1 K/decade. At the 100 hPa and cold point levels, both adjusted radiosonde data sets and reanalyses indicate large uncertainties in temperature trends. Reanalysis-based estimates at the cold point range from no trend at all (0 K/decade for ERA-Interim) to strong cooling of –1.3 K/decade (NCEP-NCAR R1). While the latter is outside of the observational uncertainty range and can be considered unrealistic, all other reanalyses data sets agree with at least one of the observational data sets within uncertainties. The bulk of the reanalyses are in good agreement at these levels, suggesting small but significant cooling trends of –0.3 K/decade to –0.6 K/decade that are statistically consistent with trends based on the adjusted radiosonde data sets.

Advances of the reanalysis and observational systems over the last decades have led to a clear improvement of the TTL reanalyses products over time. In particular, the modern reanalyses ERA-Interim, ERA5, MERRA2, CFSR and JRA-55 show a very good agreement after 2002 in terms of the vertical TTL temperature profile, meridional tropopause structure and interannual variability. Temperatures at the cold point and lapse rate, on the other hand, are too high for most old and modern reanalyses. As these differences maximise over Central Africa, a centre of deep convective activity, chemistry-transport models driven by reanalyses and simulating air mass transport into the stratosphere, can be expected to have too little dehydrations and too high water vapor. Depending on the particular application, different reanalyses offer different advantages such as a realistic cold point temperature (e.g., ERA5), small bias in the TTL temperature profile (e.g., CFSR), realistic spatial distribution of the cold point temperature (e.g., ERA-Interim), continuous TTL temperature time series through 2006 (e.g., JRA55), or a realistic representation of signals of interannual variability (e.g., MERRA2). Their use in model simulations and in comparisons with climate model output should be tailored to their specific strengths and weaknesses.

**Author contributions.** ST developed the idea for this paper and carried out the evaluations with contributions from all co-authors. SD and BL provided the reanalyses tropopause and profile data. RPK provided the GNSS-RO tropopause, wave activity and temperature profile data. JSW provided the radiosonde tropopause data. ST wrote the manuscript with contributions from all co-authors.



**Acknowledgements:**
We thank the reanalysis centres for providing their support and data products. We thank C.
Bloecker from the Global Modeling and Assimilation Office, NASA Goddard Space Flight
Center for providing information on the GNSS-RO data assimilated in MERRA-2. ERA5 data
were generated using Copernicus Climate Change Service Information. The work of S.
Tegtmeier was funded by the Deutsche Forschungsgemeinschaft (DFG, German Research
Foundation) – TE 1134/1. Contributions from J.S. Wright were supported by the National
Natural Science Foundation of China (20171352419) via a joint DFG–NSFC funding initiative.



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
