# Peer review of "Temperature and tropopause characteristics from reanalyses data in the tropical tropopause layer"

_Atmospheric Chemistry and Physics, 2019_

## Referee Comment (RC1) · Anonymous Referee #1 · 22 Jul 2019

Review of the paper:

"The tropical tropopause layer in reanalysis data sets"

written by Tegtmeyer at al.,

**General:**
This a very important and well-written paper. To understand long-term changes in the stratosphere, the tropical tropopause layer (TTL) is the most crucial region. Meteorological reanalyses are best estimates of the true state of the whole atmosphere in the past. As such, they are widely used to examine the atmospheric processes and

to detect changes in the climate system. This paper gives important insights into the representation of the TTL in all relevant reanalysis products. Thus, I would like to recommend this paper for publishing in ACP with only few minor comments and some remarks.

- General:
  I think, this is a very important statement that all reanalyses temperatures at the cold point tropopause (or at the lapse rate tropopause) show warm bias if compared to the observations because of the vertical resolution problem. Interestingly, you also show that the height of the cold point tropopause in all reanalyses is always below that derived from the observations (up to 0.4 km, Fig 6). This is an important point in the current discussion if the (tropical) deep convection is able to cross the tropopause. In many studies, water vapor and ice observations are compared with the position of the cold point tropopause derived from the reanalyses data. Because of a systematic bias of the tropopause position in the reanalyses, the observed enhanced ice/water vapor values can be erroneously attributed to transport across the tropopause. Maybe you would like to discus this point in your paper.

- General:
  In your discussion of the inter-annual variability you quantify the contribution of the QBO, volcanic eruption and linear trends. However, you do not quantify the contribution of ENSO which is also a "major player" in such variabilities. Is it because you use a zonally averaged picture and to quantify ENSO, the zonally-resolved picture would be more appropriate? If this is the case I would recommend to state this point more clearly.

- P4 L22:
  Maybe you would like to mention also more recent papers for "off-line chemistry

model applications", like Tao et al., 2019, ACP "Multitimescale variations..."

- P5, L17-23:
  I wonder, why SHADOZ data are not mentioned here which are for me still a very important tropical data set

- P6, L15:
  You explain "full-input" first in the line 41. Maybe you would like to reformulate

- P8, L12-15:
  "monthly-mean field have a warm bias of 0.5 K compared to 6-hourly data" this is not surprising. I would remove this type of motivation.

- P9, L22-23:
  "the averaged maxima and minima values" - so you count all minima and maxima and divide it by its number? How do you define a local maximum or minimum? Maybe reformulate. In any case, this procedure is important to understand Fig. 11.

- P17, L4:
  I would count "volcanic" as a tropospheric variability

- P19, L5-6:
  "During the first 15 years" - or you mean during the last 15 years (higher altitude and lower pressure - I would expect the other way around)

---

## Referee Comment (RC2) · Anonymous Referee #2 · 1 Aug 2019

This paper evaluates the vertical structure of the temperature fields from a number of meteorological reanalyses in the tropical tropopause layer (TTL). While the evaluation of reanalyses in this region is important for the user community and fits the focus of ACP, I found several limitations that should be addressed before publishing in ACP.

**General comments:**

1. From the title, I would expect that the paper also discusses wind or humidity fields in the TTL from the reanalyses, which is not the case. The title should thus be changed and I suggest "Vertical structure of temperature fields from atmospheric reanalyses in the tropical tropopause layer". Or maybe you may have a better suggestion.

[Figure]

2. I understood (Sect. 2.2) that reanalysis temperature fields in the TTL are constrained by satellite radiance observations (from 1978 onward), radiosonde profiles (from 1978 onward) and GNSS-RO (between 2002-2006 onward depending on the reanalysis). On the other hand, reanalysis temperature fields are also evaluated by radiosondes and GNSS-RO data. A proper evaluation should be done with independent datasets (i.e. not assimilated) which seems not to be the case. Please clarify and/or comment.

3. I found that the intercomparison method lack of details and/or clarity. GNSS-RO data used for the validation of the temperature are provided as zonal mean (P5L27). Is it on a daily or a monthly basis? It is also said that GNSS-RO are interpolated at the reanalysis levels (P5L35-37). A proper comparison of the reanalysis with the observations should be done by (1) mapping the reanalyses at the observation geolocation (by using additional information like averaging kernels or weighting function if necessary) to avoid sampling errors and then (2) calculating the cold point and lapse rate tropopause from the reanalyses in the space of the observations to which they are compared. If done differently, it should be justified. Please, comment and/or clarify.

4. It is said that GNSS-RO and radiosonde data are provided at high vertical resolution but their values are not given in the manuscript. Please, provide the vertical resolution of these two datasets.

5. Section 3 discusses the reanalyses between 2002 and 2010. Except that GNSS-RO data are not available before that time, is there other reasons to not show the results at earlier time? If not, I recommend providing similar figures (without GNSS-RO data) than Fig. 6 and 9 for, e.g. 1980-1990 and 1990-2000, in a supplement. This would be very instructive for the users of the reanalyses.

6. There is a long discussion about the use of model- or pressure-levels which is confusing because it seems obvious that using a low resolution standard pressure grid (only four levels in the TTL) would introduce biases. Fig. 3 is also confusing. I understand that values at 70 and 100 hPa are from the standard pressure but that the CP and LP values are calculated from the model levels. I guess that showing the temperature bias at 70 and 100 hPa from the difference profiles of Fig. 4 would provide (after interpolation) much accurate values. I would suggest to move all the discussion related to the standard pressure levels in a supplement or an appendix and to show in the main body of the paper only results obtained on the model levels

7. Both notations MERRA2 and MERRA-2 are used throughout the paper. Please, choose one of them.

**Technical corrections:**

P2 L20-22: "Model simulations..." This is not shown in the paper so it should be removed from the abstract.

P3 L11-14: "As the TTL..." Please add references at the end of the sentence.

P5 L27: "We use zonal mean..." On which time basis? Daily? Monthly? Other?

P6 L15: What do you mean by "full input"?

P6 L24-25: "MERRA-2 . . ." The meaning of this sentence is not clear. Please, clarify.

P6 L26: I would replace "produced" by "constrain" which is more accurate.

P6 L38-40: "Radiance biases..." I don't understand what message the authors want to give with this sentence. Please, clarify.

P6 L41: "...from radiosondes which..." Are these radiosonde data the same than those used for the evaluation? See also the general comment related to this issue.

P7 L10: "...from GNSS-RO instruments..." Same comment as above.

P7 L26: "While the reanalyses assimilate versions of these data..." Do you mean

"different versions of these data..."?

P7 L27: Replace "exactly" by "within their uncertainty" which is more accurate.

P7 L30-P8 L1: "In general, the..." This sentence does not describe data assimilation methodology. Instead, I suggest "Data assimilation systems combines the information from a model, a set of observations and a priori information weighted by their uncertainties."

P7 L12: I don't see the "Section 3.1" in the paper.

P9 L21: Please, add a reference to the "bootstrap method".

P9 L27-28: "The trend error..." I don't understand the meaning of this sentence. Please, clarify.

P10 L19-21: "At 100 hPa, ERA-Interim is..." I suggest redoing the figure by using different symbols (star, cross, *) allowing to see the values of all reanalyses.

P10 L22: Remove "resolution" in "...native model level resolution..."

P14 L5: I would replace "...over the Maritime continent..." by "...over the sea..." because a continent is one of the several large landmasses that make up the Earth.

P14 Figure 7: I would be very interesting to also show the results of ERA5. Is there any reason to not show it?

P15 L13: Replace "to estimating" by "to estimate".

P16 L4-5: What do you mean by "variability" in "...considerable zonal variability..."?

P16 Figure 9: Add "pressure" in the upper right panel of the figure, as in Figure 6.

P17 L17: "decrease" would be more appropriate than "improve".

P17 L29-31: "The influence of ENSO..." I do not see any figure showing the influence of ENSO on the TTL temperature. Please, clarify.

P17 L30: As explained above, change "Maritime Continent" by "sea" or "ocean".

P17 L37-P19 L5: This part is not very clear because it is never clear to which figure (10 or 11) the text refers. Please, clarify.

P17 Figure 10: Why not starting the time series in 1978 or 1980.

P21 L14-15: "...all provide realistic..." It should specify that the period of validity of this result is 2002-2010.

---

## Referee Comment (RC3) · Anonymous Referee #3 · 6 Aug 2019

General comments:

This paper evaluates the temperature structure and tropopause characteristics in the tropical tropopause layer from various meteorological reanalysis data sets. The paper is generally well written and the results of the comparison are valuable for the community. Therefore, I recommend publication after the following specific and technical comments have been addressed.

Specific comments:

1. As accurately stated in p4 L44-45, this paper investigates "key characteristics of the temperature and tropopauses in the TTL". The title, however, gives the impression that other TTL properties are also being investigated (i.e., too broad). I suggest revising

the title to indicate that the study focuses on the temperature structure and tropopause characteristics in the TTL.

2. The reasoning for choosing a certain data for certain analyses and is not always clear. Without sufficient explanation, it appears that the authors are cherry picking their results. For example:

a. Why doesn't the vertical profile for CFSR (green) in the right panel of Fig. 4 extend down to 140 hPa? Fig. 1 shows that CFSR has a model level at or just above the 140 hPa level. One of the key results, as presented in the text (e.g., Summary), is that tropical mean temperatures between 140 and 70 hPa in CFSR agrees best with those of GNSS-RO observations. I would like to see the CFSR data point near 140 hPa.

b. I would also like to see a panel using ERA5 data in Fig. 7. In all previous analyses and plots, ERA5 data are shown, but not here. Since ERA5 dataset is the newest of these reanalyses, readers will be most interested in seeing this result.

c. In Fig. 10, the temperature anomaly time series at 70 hPa (top panel) includes a time series using the RAOB radiosonde data. The second panel showing the temperature anomalies at the cold-point tropopause includes a time series using the IGRA radiosonde data. Why are the radiosonde data sources different in these two panels? Is there a reason for showing one data at 70 hPa and another at the cold point?

d. Why doesn't the right panel of Fig. 11 include data points from RATPAC, RICH and RAOBCORE (as in the left panel)?

e. The choice of radiosonde dataset in Fig. 12 is HadAT and RAOBCORE. Again, it is unclear why these two radiosonde data were chosen for this particular analysis. Perhaps it is best to stick to the same set of radiosonde data throughout the entire analyses?

3. There is a lot of discussion about the vertical resolution for obvious reasons (e.g., large impact on tropopause temperature). There is no mentioning of the horizontal

resolution of the reanalyses data used for these comparisons. While the horizontal resolution likely plays a limited role, it would be good to document what resolution was used.

Technical comments:

- p5, L20: RATPAC data are mentioned, but none of the results shown in the paper use this data.

- The second paragraph of Section 2.1 describes the various GNSS-RO measurements assimilated by the reanalyses, which are shown in Table 1. Table 1 also shows MetOp and C/NOFS data, but these are not mentioned in the text.

- p6, L32: ATOVS suite has a higher number of channels *compared to TOVS*?

- p6, L42 and p7, L12: What do you mean by "high vertical resolution"? How much higher are they compared to those of the reanalyses discussed in detail here?

- p7, L7: Is RICH also a radiosonde data (like RAOBCORE)? It is the first time this data set has been mentioned.

- p7, L5: ERA-40 reanalysis data are not analyzed in this paper. Best to leave it out?

- p8, L12: Section 3.1 does not exist. Do you mean Section 3? Or Section 2.1?

- While I see the Fig. 3 caption describing the overlapped symbols, I suggest using a different symbol so that all the data points are visible.

- It may be worthwhile to mention again at the beginning of Section 4 that the interannual variability of ERA5 variables are not analyzed due to the short data record. The sentence "In particular, . . .interannual variability" on p22, L25-28 is slightly misleading since the interannual variability in ERA5 is not analyzed.

- p17, L34: I am having difficulty seeing the positive temperature anomalies related to Mt. Pinatubo eruption in Fig. 10 (top two panels). . .

- The color of the lines for GNSS-RO and JRA-25 in Fig. 10 are difficult to distinguish. I suggest using a different color (or line style?) for JRA-25.

- Fig. 11 caption: It would be helpful to mention the 1980-2010 time period in the caption.

- p21, L31: "small negative bias at model levels *and small bias shift*, has the most realistic. . ."

---

## Referee Comment (RC4) · Anonymous Referee #4 · 6 Aug 2019

This paper evaluates data quality of multiple atmospheric reanalyses focusing on thermal characteristics of the tropical tropopause layer (TTL). The comparisons are made against long-term archives of radiosonde and GNSS RO data, which provide the most accurate temperature measurements in the TTL. Purpose of the paper is very clear, methods are reasonable, and results are well organized. It provides valuable information on reanalysis data sets and is recommended for a publication in ACP after considering several minor issues listed below.

Minor issues

1. The title is too broad. The analyses focus mainly on long-term mean features and inter-annual variability of the TTL, while the title gives an expectation that it will cover overall aspects of the TTL. Annual cycle and intra-seasonal variation are also important

features of the TTL, particularly for dehydration processes. A more detailed title is required if authors decide not to include these features. One suggestion is making this paper as "part 1" covering long-term structures and inter-annual variability and left annual cycle and intra-seasonal variability for a future study (as this paper already has enough material, I think...).

2. This is also related to comment #1. The temperature bias peaking near the equator and its potential connection to Kelvin waves (Figs. 6-8) are interesting results. This part is worth to be further investigated (even in a different paper) as it provides noble information for researchers studying the dehydration process based on reanalyses. Particularly, this feature could be "seasonally" different because temperature and circulation structures in the TTL undergo strong seasonality. The same is true for the Kelvin source over central Africa.

3. Please provide some details describing how the CPT/LRT and their properties are calculated in this study. Several methods have been used to estimate properties of the CPT, and the results could be sensitive to the selected method, particularly for data set with coarse vertical resolution. This information will be helpful for readers to better understand the results provided in this paper.

4. Given the accuracy and vertical resolution of ERA5 described in section 3, CPT temperature trend from ERA5 would be most reliable. It will be very useful if this information could be added in Fig. 12. (just suggestion)

5. Dynamical aspect (e.g., upwelling) in the TTL is not covered in this paper. Some discussion may be beneficial (but not necessary).

Technical comments

P3L20: Pan and Munchak (2011), Pan et al. (2018) could be good references for this paragraph P3L37: 0.5 km is roughly 5 hPa at this level, 5 hPa maybe more consistent P10L29: "near 100 hPa (ERA-Interim; -0.82 K)". This is correct in Fig. 4 at ∼96 hPa, but

could look inconsistent with Fig. 3 (right panel) as it shows ∼ -0.4 K at 100 hPa. Better to mention that it is on a model level, not 100 hPa. Fig. 4: Average on pressure level could be a bit misleading as it shows a smooth CPT. Additional figure on tropopause relative coordinate (e.g., Birner et al. 2002) could be useful. P12L1: "comes at the expense . . . tropopause". This expression could be a bit misleading because there is no clear causality. P13L5: "with respect to the zonal mean" => in meridional direction? Fig. 7: ERA5 could provide an important clue on this issue as it has a good vertical resolution, but it is missing in the figure. Fig. 8: Is the left figure different from that in Fig. 7? P17L24: datasets => data sets Fig. 10: RAOB is used for the first figure, but IGRA is used for the second figure. It will be helpful if an explanation is provided why authors made this choice. Periods ('.') are missing in several section titles and figure captions.

References

Birner, T., A. Dornbrack, and U. Schumann, 2002: How sharp is the tropopause at midlatitudes? Geophys. Res. Lett., 29, 1700. Pan, L. L., and L. a. Munchak. (2011). Relationship of cloud top to the tropopause and jet structure from CALIPSO data. J. Geophys. Res., 116, D12201. Pan, L. L., Honomichl, S. B., Bui, T. V., Thornberry, T., Rollins, A., Hintsa, E., & Jensen, E. J. (2018). Lapse Rate or Cold Point: The Tropical Tropopause Identified by In Situ Trace Gas Measurements. Geophysical Research Letters, 45(19), 10-756.

―――――――――――――――――

---

## Author Comment (AC1) · 24 Oct 2019

**Response to interactive comments on "The tropical tropopause layer in reanalysis data sets" by Tegtmeier et al.**

We thank the reviewer for his/her comments which have helped us to improve the paper in revision. Comments are reproduced below, followed by our responses in *italics*.

**Anonymous Referee #1**

General:
This a very important and well-written paper. To understand long-term changes in the stratosphere, the tropical tropopause layer (TTL) is the most crucial region. Meteorological reanalyses are best estimates of the true state of the whole atmosphere in the past. As such, they are widely used to examine the atmospheric processes and to detect changes in the climate system. This paper gives important insights into the representation of the TTL in all relevant reanalysis products. Thus, I would like to recommend this paper for publishing in ACP with only few minor comments and some remarks.

General:
I think, this is a very important statement that all reanalyses temperatures at the cold point tropopause (or at the lapse rate tropopause) show warm bias if compared to the observations because of the vertical resolution problem. Interestingly, you also show that the height of the cold point tropopause in all reanalyses is always below that derived from the observations (up to 0.4 km, Fig 6). This is an important point in the current discussion if the (tropical) deep convection is able to cross the tropopause. In many studies, water vapor and ice observations are compared with the position of the cold point tropopause derived from the reanalyses data. Because of a systematic bias of the tropopause position in the reanalyses, the observed enhanced ice/water vapor values can be erroneously attributed to transport across the tropopause. Maybe you would like to discuss this point in your paper.

> *Thanks for pointing this out. This is indeed an interesting implication of the tropopause altitude comparison. We have added a statement to the summary.*

In your discussion of the inter-annual variability you quantify the contribution of the QBO, volcanic eruption and linear trends. However, you do not quantify the contribution of ENSO which is also a "major player" in such variabilities. Is it because you use a zonally averaged picture and to quantify ENSO, the zonally-resolved picture would be more appropriate? If this is the case, I would recommend to state this point more clearly.

> *Yes, including ENSO in the zonally averaged multilinear regression study does not allow for conclusive results as the zonally varying ENSO signals cancel each other out in the zonal mean analyses. We have also conducted multilinear regression of the zonally resolved temperature fields that will be discussed in a follow up publication, currently in preparation. The manuscript contains a statement explaining this ' … The influence of ENSO on TTL temperatures shows large longitudinal variations with positive anomalies over the Maritime Continent and West Pacific and negative anomalies over the East Pacific. While the zonally resolved response patterns agree well between observations and reanalyses (not shown here), the zonal mean responses are not significant (not shown here) …'.*

P4 L22: Maybe you would like to mention also more recent papers for "off-line chemistry model applications", like Tao et al., 2019, ACP "Multitimescale variations..."

*We have added the reference to the manuscript.*

P5, L17-23: I wonder, why SHADOZ data are not mentioned here which are for me still a very important tropical data set

*We have not used SHADOZ data, as this record (starting in 1999) is not long enough for the comparison of interannual variability and long-term changes evaluated here for the S-RIP core time period (1980–2010). For the zonal mean climatological analyses of the time period after 2000 we decided to use the GNSS-RO data as their uniform horizontal coverage allows to include tropical and zonal mean comparisons.*

P6, L15: You explain "full-input" first in the line 41. Maybe you would like to reformulate

*We use the term "full-input" reanalyses here for systems that assimilate surface and upper-air conventional and satellite data (compared to systems that only assimilate surface observations). This information is given in line 15. We slightly reformulated the sentence to make this clearer.*

P8, L12-15: "monthly-mean field have a warm bias of 0.5 K compared to 6-hourly data" this is not surprising. I would remove this type of motivation.

*We have removed this sentence from the manuscript.*

P9, L22-23: "the averaged maxima and minima values" - so you count all minima and maxima and divide it by its number? How do you define a local maximum or minimum? Maybe reformulate. In any case, this procedure is important to understand Fig. 11.

*We have added the following information to the manuscript '… For each QBO cycle of this time series, the absolute temperature maximum and minimum are selected. In a second step, the means over all such temperature maxima and minima are calculated to give the averaged maximum and minimum values, respectively …'*

P17, L4: I would count "volcanic" as a tropospheric variability

*As the positive temperature anomalies in the upper TTL associated with volcanic eruptions are related to volcanic stratospheric aerosols, we have decided to list volcanic here under stratospheric variability.*

P19, L5-6: "During the first 15 years" - or you mean during the last 15 years (higher altitude and lower pressure - I would expect the other way around)

*Thanks for pointing this out. We refer here to the first 15 years and the wording was mixed up. The sentence has been corrected in the manuscript.*

---

## Author Comment (AC2) · 24 Oct 2019

**Response to interactive comments on "The tropical tropopause layer in reanalysis data sets" by Tegtmeier et al.**

We thank the reviewer for his/her comments which have helped us to improve the paper in revision. Comments are reproduced below, followed by our responses in *italics*.

**Anonymous Referee #2**

This paper evaluates the vertical structure of the temperature fields from a number of meteorological reanalyses in the tropical tropopause layer (TTL). While the evaluation of reanalyses in this region is important for the user community and fits the focus of ACP, I found several limitations that should be addressed before publishing in ACP.

General comments:
1. From the title, I would expect that the paper also discusses wind or humidity fields in the TTL from the reanalyses, which is not the case. The title should thus be changed and I suggest "Vertical structure of temperature fields from atmospheric reanalyses in the tropical tropopause layer". Or maybe you may have a better suggestion.

*We agree that the title was too broad and have changed it to 'Temperature and tropopause characteristics from atmospheric reanalyses in the tropical tropopause layer'.*

2. I understood (Sect. 2.2) that reanalysis temperature fields in the TTL are constrained by satellite radiance observations (from 1978 onward), radiosonde profiles (from 1978 onward) and GNSS-RO (between 2002-2006 onward depending on the reanalysis). On the other hand, reanalysis temperature fields are also evaluated by radiosondes and GNSS-RO data. A proper evaluation should be done with independent datasets (i.e. not assimilated) which seems not to be the case. Please clarify and/or comment.

*We agree that ideally an evaluation would be based on independent data sets. Unfortunately, there is no independent temperature data set with the required spatial coverage, uniform sampling and vertical resolution available in the TTL region.*

3. I found that the intercomparison method lack of details and/or clarity. GNSS-RO data used for the validation of the temperature are provided as zonal mean (P5L27). Is it on a daily or a monthly basis? It is also said that GNSS-RO are interpolated at the reanalysis levels (P5L35-37). A proper comparison of the reanalysis with the observations should be done by (1) mapping the reanalyses at the observation geolocation (by using additional information like averaging kernels or weighting function if necessary) to avoid sampling errors and then (2) calculating the cold point and lapse rate tropopause from the reanalyses in the space of the observations to which they are compared. If done differently, it should be justified.

*The intercomparison of GNSS-RO data to reanalyses model level temperature (e.g., Figure 4 and 5) is based on the following method. For each individual profile the temperature is interpolated from the two adjacent levels to the reanalyses model level based on the barometric formula. In a second step, the monthly mean tropical mean values are calculated.*
*The intercomparison of GNSS-RO data to reanalyses cold point and lapse rate tropopause (e.g., Figure 3, 5 and 6) is based on the following method. For each profile, the cold point*

*and lapse rate tropopause characteristics were identified based on the cold point and WMO criteria, respectively. In a second step, the monthly mean zonal mean and monthly mean tropical mean values are calculated.*

*Zonal averages of GNSS-RO data do not suffer from uneven sampling patterns as they are evenly distributed over longitude on a monthly basis (see Fig. 3 of Yu, K., Rizos, C., Burrage, D. et al., An overview of GNSS remote sensing, EURASIP J. Adv. Signal Process. (2014) 134. https://doi.org/10.1186/1687-6180-2014-134).*

*We have added information to section 2.1 of the manuscript to explain the methodology in more detail.*

Please, comment and/or clarify.

4. It is said that GNSS-RO and radiosonde data are provided at high vertical resolution but their values are not given in the manuscript. Please, provide the vertical resolution of these two datasets.

> *We have added the following information to the manuscript* 'The GNSS-RO 'wetPrf' temperature profiles from CDAAC are provided on a 100-m vertical grid from the surface to 40 km altitude. The effective physical resolution is variable, ranging from ~1 km in regions of constant stratification down to 100-200m where the biggest stratification gradients occur e.g. at the top of the boundary layer or at a very sharp tropopause (Kursinski et al., 1997; Gorbunov et al., 2004), most often being somewhere in between.'
>
> *Regarding the vertical resolution of radiosondes, in addition to mandatory levels (which near the tropical tropopause are 150, 100, 70, and 50 hPa), individual radiosonde soundings include data at "significant levels," where the observations between mandatory reporting levels depart from a linear interpolation, such as would occur at the tropopause. As the number of significant levels can vary over time and with station, a conclusive statement on the vertical resolution is not possible. We have therefore removed 'high-resolution' from the sentence.*

5. Section 3 discusses the reanalyses between 2002 and 2010. Except that GNSS-RO data are not available before that time, is there other reasons to not show the results at earlier time? If not, I recommend providing similar figures (without GNSS-RO data) than Fig. 6 and 9 for, e.g. 1980-1990 and 1990-2000, in a supplement. This would be very instructive for the users of the reanalyses.

> *We have added the zonal mean evaluations of the lapse rate and cold point tropopause for the time periods 1981-1990 and 1991-2000 in a supplement.*

6. There is a long discussion about the use of model- or pressure-levels which is confusing because it seems obvious that using a low-resolution standard pressure grid (only four levels in the TTL) would introduce biases. Fig. 3 is also confusing. I understand that values at 70 and 100 hPa are from the standard pressure but that the CP and LP values are calculated from the model levels. I guess that showing the temperature bias at 70 and 100 hPa from the difference profiles of Fig. 4 would provide (after interpolation) much accurate values. I would suggest to move all the discussion related to the standard pressure levels in a supplement or an appendix and to show in the main body of the paper only results obtained on the model levels.

> *We agree with the reviewer, that it is not surprising that the low-resolution standard levels introduce biases when used for tropopause calculations. Our sensitivity test is used to illustrate how large such a bias can be for the tropopause temperature, altitude and*

*pressure calculations. We have simplified and shortened the discussion of this issue, to make this point clearer.*

*Figure 3 uses the 70 and 100 hPa levels to present the comparison for all reanalyses at the same level. This temperature comparison on pressure levels offers additional information to the comparison on model levels presented in Figure 4. This additional information is valuable for studies that have or will use pressure levels instead of model levels in the TTL regions.*

7. Both notations MERRA2 and MERRA-2 are used throughout the paper. Please, choose one of them.

*We have changed the notations to MERRA-2.*

Technical corrections:

P2 L20-22: "Model simulations. . ." This is not shown in the paper so it should be removed from the abstract.

*We have removed the sentence from the abstract.*

P3 L11-14: "As the TTL. . ." Please add references at the end of the sentence.

*We have added three references to the sentence.*

P5 L27: "We use zonal mean. . ." On which time basis? Daily? Monthly? Other?

*We have added the information 'monthly mean' to the sentence.*

P6 L15: What do you mean by "full input"?

*We use the term "full-input" reanalyses here for systems that assimilate surface and upper-air conventional and satellite data (compared to systems that only assimilate surface observations). We have reformulated the sentence to make this clearer.*

P6 L24-25: "MERRA-2 . . ." The meaning of this sentence is not clear. Please, clarify.

*We have moved the sentence to the acknowledgements.*

P6 L26: I would replace "produced" by "constrain" which is more accurate.

*We have changed the wording as suggested.*

P6 L38-40: "Radiance biases. . ." I don't understand what message the authors want to give with this sentence. Please, clarify.

*We have replace the sentence with '… Because radiance biases associated with instrument changes, inaccurate calibration offsets, orbital drifts or long-term $CO_2$ changes can cause unwanted biases in the resulting reanalysis temperature fields (e.g. Rienecker et al., 2011), a variational bias correction scheme is used during the data assimilation procedure to remove or minimize any radiance biases. This ensures that any temperature changes introduced by the circumstances outlined above are kept small, which is important when looking for long term changes. …'.*

P6 L41: ". . .from radiosondes which. . ." Are these radiosonde data the same than those used for the evaluation? See also the general comment related to this issue.

*To a large degree the assimilated radiosonde data profiles are the same as the ones used for the homogenized radiosonde data sets. As each radiosonde data set uses different*

*criteria on which stations and profiles to include, there exist small differences between assimilated and homogenized radiosonde data sets.*

*Also not the sentence later in the paragraph* 'In order to avoid discontinuities or inconsistencies in temperature time series from radiosondes, several reanalysis systems use homogenized temperature data sets such as RAOBCORE (ERA-Interim, JRA-55, MERRA, MERRA-2) and RICH (ERA5).'.

P7 L10: ". . .from GNSS-RO instruments. . ." Same comment as above.
*The GNSS-RO instruments assimilated by the reanalyses are the same used for the evaluation of the data sets. We have added the sentence '* ... In addition to the GNSS-RO data sets discussed in section 2.1, C/NOFS-CORISS (Communications/Navigation Outage Forecasting System Occultation Receiver for Ionospheric Sensing and Specification) is assimilated by some of the reanalyses.' *to make this clearer.*

P7 L26: "While the reanalyses assimilate versions of these data..." Do you mean "different versions of these data. . ."?
*We have added 'different' to the sentence.*

P7 L27: Replace "exactly" by "within their uncertainty" which is more accurate.
*We have changed the sentence accordingly.*

P7 L30-P8 L1: "In general, the. . ." This sentence does not describe data assimilation methodology. Instead, I suggest "Data assimilation systems combines the information from a model, a set of observations and a priori information weighted by their uncertainties."
*We have changed the sentence accordingly.*

P7 L12: I don't see the "Section 3.1" in the paper.
*We have changed the text to '*Section 2.3'.

P9 L21: Please, add a reference to the "bootstrap method".
*We have added the reference:* Efron, B., and R. J. Tibshirani (1993), An Introduction to the Bootstrap, 436 pp., Chapman and Hall, New York.

P9 L27-28: "The trend error..." I don't understand the meaning of this sentence. Please, clarify.
*We have changed the sentence to '*... The uncertainty in each long-term trend is calculated as the standard error of the slope with the effective sample size adjusted to account for the corresponding lag-1 autocorrelation coefficient. ...'

P10 L19-21: "At 100 hPa, ERA-Interim is..." I suggest redoing the figure by using different symbols (star, cross, *) allowing to see the values of all reanalyses.
*We have produced different versions of this figure (including different symbols or symbols slightly shifted vertically), but found that the visibility does not improve sufficiently. Therefore, we prefer to keep the figure in its current version and to mention the overlaps in the caption.*

P10 L22: Remove "resolution" in ". . .native model level resolution. . ."
*We have changed the text accordingly.*

P14 L5: I would replace "...over the Maritime continent..." by "...over the sea..." because a continent is one of the several large landmasses that make up the Earth.

*As the expression 'maritime continent' has been used in many TTL publications to refer to the overall region including landmasses and sea, we prefer to keep the expression. We will follow the example from Fueglistaler et al. (2009) and use maritime continent in quotes at the first occurrence.*

P14 Figure 7: I would be very interesting to also show the results of ERA5. Is there any reason to not show it?

*We have added the latitude–longitude comparison of cold point temperature for ERA5 to Figure 7.*

P15 L13: Replace "to estimating" by "to estimate".

*We have changed the text accordingly.*

P16 L4-5: What do you mean by "variability" in ". . .considerable zonal variability. . ."?

*We have changed the sentence to ' … The altitude of the lapse rate tropopause shows considerable meridional variability, ranging from 14.5 km to 16.7 km. …'.*

P16 Figure 9: Add "pressure" in the upper right panel of the figure, as in Figure 6.

*We have added the label 'pressure' to the panel.*

P17 L17: "decrease" would be more appropriate than "improve".

*We have changed the sentence accordingly.*

P17 L29-31: "The influence of ENSO. . ." I do not see any figure showing the influence of ENSO on the TTL temperature. Please, clarify.

*As we focus here on the zonal mean interannual variability, we do not show the longitudinal temperature variations associated with the ENSO signal. We have moved the phrase 'not shown here' from the next sentence to this sentence, to make this clear from the onset.*

P17 L30: As explained above, change "Maritime Continent" by "sea" or "ocean".

*As the expression 'maritime continent' has been used in many TTL publications to refer to the overall region including landmasses and sea, we prefer to keep the expression. We will follow the example from Fueglistaler et al. (2009) and use maritime continent in quotes at the first occurrence.*

P17 L37-P19 L5: This part is not very clear because it is never clear to which figure (10 or 11) the text refers. Please, clarify.

*The text refers to Figure 10, except for the last sentence. We have added this information.*

P17 Figure 10: Why not starting the time series in 1978 or 1980.

*For consistency with the S-RIP report and other publications, we use here the S-RIP climatological core time period January 1981 to December 2010.*

P21 L14-15: "...all provide realistic..." It should specify that the period of validity of this result is 2002-2010.

*We have changed the sentence accordingly.*

---

## Author Comment (AC4) · 24 Oct 2019

**Response to interactive comments on "The tropical tropopause layer in reanalysis data sets" by Tegtmeier et al.**

We thank the reviewer for his/her comments which have helped us to improve the paper in revision. Comments are reproduced below, followed by our responses in *italics*.

**Anonymous Referee #3**

General comments:
This paper evaluates the temperature structure and tropopause characteristics in the tropical tropopause layer from various meteorological reanalysis data sets. The paper is generally well written and the results of the comparison are valuable for the community. Therefore, I recommend publication after the following specific and technical comments have been addressed.

Specific comments:
1. As accurately stated in p4 L44-45, this paper investigates "key characteristics of the temperature and tropopauses in the TTL". The title, however, gives the impression that other TTL properties are also being investigated (i.e., too broad). I suggest revising the title to indicate that the study focuses on the temperature structure and tropopause characteristics in the TTL.

> *We agree that the title was too broad and have changed it to 'Temperature and tropopause characteristics from atmospheric reanalyses in the tropical tropopause layer'.*

2. The reasoning for choosing a certain data for certain analyses and is not always clear. Without sufficient explanation, it appears that the authors are cherry picking their results. For example:
a. Why doesn't the vertical profile for CFSR (green) in the right panel of Fig. 4 extend down to 140 hPa? Fig. 1 shows that CFSR has a model level at or just above the 140 hPa level. One of the key results, as presented in the text (e.g., Summary), is that tropical mean temperatures between 140 and 70 hPa in CFSR agrees best with those of GNSS-RO observations. I would like to see the CFSR data point near 140 hPa.

> *Indeed, the CFSR model level at around 138 hPa should be included in this evaluation. We have added CFSR at this level to Figure 4. Results remain unchanged as CFSR temperature at the model level around 138 hPa also agrees very well with the GNSS-RO data.*

b. I would also like to see a panel using ERA5 data in Fig. 7. In all previous analyses and plots, ERA5 data are shown, but not here. Since ERA5 dataset is the newest of these reanalyses, readers will be most interested in seeing this result.

> *We have added the latitude–longitude comparison of cold point temperature for ERA5 to Figure 7. It shows a structure very similar to the other reanalyses when compared to the observations.*

c. In Fig. 10, the temperature anomaly time series at 70 hPa (top panel) includes a time series using the RAOB radiosonde data. The second panel showing the temperature anomalies at the cold-point tropopause includes a time series using the IGRA radiosonde data. Why are the

radiosonde data sources different in these two panels? Is there a reason for showing one data at 70 hPa and another at the cold point?

> *For consistency reasons we decided to rely on the radiosonde data sets used in Wang et al., 2013, where the authors provide detailed evaluations of the temporal variability and trends of radiosonde temperature in the TTL. Wang et al. (2013) use the unadjusted quality-controlled radiosonde data set IGRA for the cold point and several independently adjusted radiosonde temperature data sets RATPAC, HadAT, RAOBCORE, and RICH for temperatures at 70 and 100 hPa. The motivation for evaluating interannual variability of cold point temperature, height and pressure only from the unadjusted temperature profiles is that temperature adjustments can change the location of the cold point tropopause in a profile. Therefore, we show RAOBCORE in the top panel at 70 hPa and IGRA in the lower panels for the cold point. The interannual anomalies at 70 hPa are shown only for RAOBCORE for a better clarity of the figure, while the other data sets are mentioned in the text. We have added a detailed explanation to chapter 2.1 (Observational data sets) to make clear which data sets are used at which levels for which reasons.*

d. Why doesn't the right panel of Fig. 11 include data points from RATPAC, RICH and RAOBCORE (as in the left panel)?

> *Same reason as above.*

e. The choice of radiosonde dataset in Fig. 12 is HadAT and RAOBCORE. Again, it is unclear why these two radiosonde data sets were chosen for this particular analysis. Perhaps it is best to stick to the same set of radiosonde data throughout the entire analyses?

> *For the trends at 70 and 100 hPa, we show the smallest and largest trends derived from the four adjusted radiosonde data sets as reported in Wang et al. (2012) and consider their range (including the reported error bars) as the observational uncertainty range. We have added this information to the Methods section.*

3. There is a lot of discussion about the vertical resolution for obvious reasons (e.g., large impact on tropopause temperature). There is no mentioning of the horizontal resolution of the reanalyses data used for these comparisons. While the horizontal resolution likely plays a limited role, it would be good to document what resolution was used.

> *We have added information on the horizontal resolutions of the reanalyses data sets.*

Technical comments:
- p5, L20: RATPAC data are mentioned, but none of the results shown in the paper use this data.

> *RATPAC results are shown in Figure 11.*

- The second paragraph of Section 2.1 describes the various GNSS-RO measurements assimilated by the reanalyses, which are shown in Table 1. Table 1 also shows MetOp and C/NOFS data, but these are not mentioned in the text.

> *We have added the information to the text.*

- p6, L32: ATOVS suite has a higher number of channels *compared to TOVS*?

> *We have changed the sentence accordingly.*

- p6, L42 and p7, L12: What do you mean by "high vertical resolution"? How much higher are they compared to those of the reanalyses discussed in detail here?

> *We have added the following information to the manuscript* 'The GNSS-RO 'wetPrf' temperature profiles from CDAAC are provided on a 100-m vertical grid from the surface to 40 km altitude. The effective physical resolution is variable, ranging from ~1 km in regions of constant stratification down to 100-200m where the biggest stratification gradients occur e.g. at the top of the boundary layer or at a very sharp tropopause (Kursinski et al., 1997; Gorbunov et al., 2004), most often being somewhere in between.'
>
> *Regarding the vertical resolution of radiosondes, in addition to mandatory levels (which near the tropical tropopause are 150, 100, 70, and 50 hPa), individual radiosonde soundings include data at "significant levels," where the observations between mandatory reporting levels depart from a linear interpolation, such as would occur at the tropopause. As the number of significant levels can vary over time and with station, a conclusive statement on the vertical resolution is not possible. We have therefore removed 'high-resolution' from the sentence.*

- p7, L7: Is RICH also a radiosonde data (like RAOBCORE)? It is the first time this data set has been mentioned.

> *We have added RICH to Section 2.*

- p7, L5: ERA-40 reanalysis data are not analyzed in this paper. Best to leave it out?

> *We have now included ERA-40 in two supplementary figures covering earlier time periods and therefore retain this text.*

- p8, L12: Section 3.1 does not exist. Do you mean Section 3? Or Section 2.1?

> *We have changed the text to section 2.3.*

- While I see the Fig. 3 caption describing the overlapped symbols, I suggest using a different symbol so that all the data points are visible.

> *We have produced different versions of this figure (including different symbols or symbols slightly shifted vertically), but found that the visibility does not improve sufficiently. Therefore, we prefer to keep the figure in its current version and to mention the overlaps in the caption.*

- It may be worthwhile to mention again at the beginning of Section 4 that the interannual variability of ERA5 variables are not analyzed due to the short data record. The sentence "In particular, . . .interannual variability" on p22, L25-28 is slightly misleading since the interannual variability in ERA5 is not analyzed.

> *We have added this information to the beginning of Section 4. We change the sentence on page 22 to* 'In particular, the more recent reanalyses ERA-Interim, ERA5, MERRA-2, CFSR and JRA-55 mostly show very good agreement after 2002 in terms of the vertical TTL temperature profile, meridional tropopause structure and interannual variability.'

- p17, L34: I am having difficulty seeing the positive temperature anomalies related to Mt. Pinatubo eruption in Fig. 10 (top two panels).

> *Thanks for pointing this out. It is true, that following Mount Pinatubo only weak temperature anomalies occur at 70 hPa and no anomalies occur at the cold point. This is consistent with Fujiwara et al. (2015) who show that the positive temperature anomalies*

*following Mount Pinatubo do not propagate down as far into the TTL as the ones following El Chichón. We have changed the text accordingly.*

- The color of the lines for GNSS-RO and JRA-25 in Fig. 10 are difficult to distinguish. I suggest using a different color (or line style?) for JRA-25.
  *We have decided to keep the colors used for the reanalyses consistent with the S-RIP colour scheme. We have changed the color used for GNSS-RO to a slightly darker grey to make it more distinguishable from JRA-25.*

- Fig. 11 caption: It would be helpful to mention the 1980-2010 time period in the caption.
  *We have added the time period to the caption.*

- p21, L31: "small negative bias at model levels *and small bias shift*, has the most realistic."
  *We have added 'and a small bias shift' to the text.*

---

## Author Comment (AC5) · 24 Oct 2019

**Response to interactive comments on "The tropical tropopause layer in reanalysis data sets" by Tegtmeier et al.**

We thank the reviewer for his/her comments which have helped us to improve the paper in revision. Comments are reproduced below, followed by our responses in *italics*.

**Anonymous Referee #4**

This paper evaluates data quality of multiple atmospheric reanalyses focusing on ther- mal characteristics of the tropical tropopause layer (TTL). The comparisons are made against long-term archives of radiosonde and GNSS RO data, which provide the most accurate temperature measurements in the TTL. Purpose of the paper is very clear, methods are reasonable, and results are well organized. It provides valuable information on reanalysis data sets and is recommended for a publication in ACP after considering several minor issues listed below.

Minor issues
1. The title is too broad. The analyses focus mainly on long-term mean features and inter-annual variability of the TTL, while the title gives an expectation that it will cover overall aspects of the TTL. Annual cycle and intra-seasonal variation are also features of the TTL, particularly for dehydration processes. A more detailed title is required if authors decide not to include these features. One suggestion is making this paper as "part 1" covering long-term structures and inter-annual variability and left annual cycle and intra-seasonal variability for a future study (as this paper already has enough material, I think. . .).

> *We agree that the title was too broad and have changed it to 'Temperature and tropopause characteristics from atmospheric reanalyses in the tropical tropopause layer'. We make it clear from the onset that we focus on climatological and long-term characteristics by adding this information at the beginning of the abstract.*

2. This is also related to comment #1. The temperature bias peaking near the equator and its potential connection to Kelvin waves (Figs. 6-8) are interesting results. This part is worth to be further investigated (even in a different paper) as it provides noble information for researchers studying the dehydration process based on reanalyses. Particularly, this feature could be "seasonally" different because temperature and circulation structures in the TTL undergo strong seasonality. The same is true for the Kelvin source over central Africa.

> *We agree with the reviewer that the evaluation of the seasonal cycle would be an interesting addition and should be covered in a future follow-up study. We have added a remark to the summary section.*

3. Please provide some details describing how the CPT/LRT and their properties are calculated in this study. Several methods have been used to estimate properties of the CPT, and the results could be sensitive to the selected method, particularly for data set with coarse vertical resolution. This information will be helpful for readers to better understand the results provided in this paper.

> *We have added the following text to the manuscript* 'We derive the cold point and lapse rate tropopause characteristics for each reanalysis using model-level data between 500 and 10 hPa at each grid point at 6-hourly temporal resolution. Zonal and long-term averages are calculated by averaging over all grid points, and represent the final step of data processing.

For our calculations, the cold point tropopause is defined as the coldest model level. The lapse rate tropopause is defined as the lowest level at which the lapse rate decreases to 2 K km$^{-1}$ or less, provided that the average lapse rate between this level and all higher levels within 2 km does not exceed 2 K km$^{-1}$ (World Meteorological Organization, 1957).'

4. Given the accuracy and vertical resolution of ERA5 described in section 3, CPT temperature trend from ERA5 would be most reliable. It will be very useful if this information could be added in Fig. 12. (just suggestion)

*At the moment, we have only acquired the 6 hourly data sets from ERA5 (used to calculated the cold point tropopause) for the period 2000 to 2017. We can also access a data set on the 37 standard CFMIP pressure levels for the time period 1979 to 2017, but the lower vertical resolution will impact the cold point tropopause. Therefore, we have decided to include long-term changes of ERA5 in a follow-up study that focuses on a detailed comparison of ERA-Interim and ERA5 and will evaluate long-term changes over different time periods (i.e., extending to 2018).*

5. Dynamical aspect (e.g., upwelling) in the TTL is not covered in this paper. Some discussion may be beneficial (but not necessary).

*We agree that a discussion of the dynamical aspects of the TTL would very interesting. However, this would require a large amount of additional material and is beyond the scope of the manuscript. Such a discussion can be found in the TTL chapter of the upcoming S-RIP report (currently under review) and related paper publications.*

Technical comments
P3L20: Pan and Munchak (2011), Pan et al. (2018) could be good references for this paragraph
*We have added the references to the manuscript.*

P3L37: 0.5 km is roughly 5 hPa at this level, 5 hPa maybe more consistent.
*Thanks for pointing this out. We have changed the text to 7 hPa.*

P10L29: "near 100 hPa (ERA-Interim; -0.82 K)". This is correct in Fig. 4 at ~96 hPa, but could look inconsistent with Fig. 3 (right panel) as it shows ~ -0.4 K at 100 hPa. Better to mention that it is on a model level, not 100 hPa.
*We have changed the text according to the suggestions.*

Fig. 4: Average on pressure level could be a bit misleading as it shows a smooth CPT. Additional figure on tropopause relative coordinate (e.g., Birner et al. 2002) could be useful.
*We have decided to keep the current version of Figure 4, which aims at identifying reanalysis biases on the respective model levels. Tropopause-based averages would have the disadvantage to be affected by cold point altitude biases. Therefore, biases in a tropopause-based figure would belong to different model levels mixed up in the same tropopause-relative level.*

P12L1: "comes at the expense . . . tropopause". This expression could be a bit misleading because there is no clear causality.
*We have changed the sentence to phrase this more carefully.*

P13L5: "with respect to the zonal mean" => in meridional direction?

*Yes, this phrase fits better. We have changed the sentence.*

Fig. 7: ERA5 could provide an important clue on this issue as it has a good vertical resolution, but it is missing in the figure.

*We have added ERA5 to the figure. It shows very similar structures when compared to the observations.*

Fig. 8: Is the left figure different from that in Fig. 7?

*Only the colour bar is different in order to contrast the comparison of CFSR over the whole time period better with the comparison during times of high Kelvin wave activity.*

P17L24: datasets => data sets Fig. 10: RAOB is used for the first figure, but IGRA is used for the second figure. It will be helpful if an explanation is provided why authors made this choice. Periods ('.') are missing in several section titles and figure captions.

*For consistency reasons we decided to rely on the radiosonde data sets used in Wang et al., 2013, where the authors provide detailed evaluations of the temporal variability and trends of radiosonde temperature in the TTL. Wang et al. (2013) use the unadjusted quality-controlled radiosonde data set IGRA for the cold point and several independently adjusted radiosonde temperature data sets RATPAC, HadAT, RAOBCORE, and RICH for temperatures at 70 and 100 hPa. The motivation for evaluating interannual variability of cold point temperature, height and pressure only from the unadjusted temperature profiles is that temperature adjustments can change the location of the cold point tropopause in a profile. Therefore, we show RAOBCORE in the top panel at 70 hPa and IGRA in the lower panels for the cold point. The interannual anomalies at 70 hPa are shown only for RAOBCORE for a better clarity of the figure, while the other data sets are mentioned in the text. We have added a detailed explanation to chapter 2.1 (Observational data sets) to make clear which data sets are used at which levels for which reasons.*

References

Birner, T., A. Dornbrack, and U. Schumann, 2002: How sharp is the tropopause at midlatitudes? Geophys. Res. Lett., 29, 1700.

Pan, L. L., and L. a. Munchak. (2011). Relationship of cloud top to the tropopause and jet structure from CALIPSO data. J. Geophys. Res., 116, D12201.

Pan, L. L., Honomichl, S. B., Bui, T. V., Thornberry, T., Rollins, A., Hintsa, E., & Jensen, E. J. (2018). Lapse Rate or Cold Point: The Tropical Tropopause Identified by In Situ Trace Gas Measurements. Geophysical Research Letters, 45(19), 10-756.

---

## Author Response (AR2)

**Response to interactive comments on "The tropical tropopause layer in reanalysis data sets" by Tegtmeier et al.**

We thank the reviewer for his/her comments which have helped us to improve the paper in revision. Comments are reproduced below, followed by our responses in *italics*. Changes to the manuscript are marked in blue bold print.

**Anonymous Referee #2**

I wish to thank the authors to have considered my previous comments and to have addressed them positively. I still have two (minor) comments and I suggest publishing the paper after having considered them.

1.The reanalyses discussed in this paper are introduced too far away (in Sect. 2.2). Their acronyms are also undefined. I suggest to introduce the reanalyses in the introduction (including the definition of the acronym) and, possibly also, in the abstract. In the abstract, I would also provide the name of the reanalyses concerned by "most recent atmospheric reanalysis …" (P2L12).

*We have added a list of the reanalyses discussed in the paper to the introduction and abstract. Acronyms are defined in section 2.2.*

2. According to Sherwood (2000, GRL), the "maritime continent" corresponds to the general area of Indonesia. I thus suggest to replace (P15L5) "… "maritime continent"… " by "… "
[revised manuscript text omitted]